# RETRIEVAL-AUGMENTED DECISION TRANSFORMER: EXTERNAL MEMORY FOR IN-CONTEXT RL

## ABSTRACT

In-context learning (ICL) is the ability of a model to learn a new task by observing a few exemplars in its context. While prevalent in NLP, this capability has recently also been observed in Reinforcement Learning (RL) settings. Prior in-context RL methods, however, require entire episodes in the agent's context. Given that complex environments typically lead to long episodes with sparse rewards, these methods are constrained to simple environments with short episodes. To address these challenges, we introduce Retrieval-Augmented Decision Transformer (RA-DT). RA-DT employs an external memory mechanism to store past experiences from which it retrieves only sub-trajectories relevant for the current situation. The retrieval component in RA-DT does not require training and can be entirely domain-agnostic. We evaluate the capabilities of RA-DT on grid-world environments, robotics simulations, and procedurally-generated video games. On grid-worlds, RA-DT outperforms baselines, while using only a fraction of their context length. Furthermore, we illuminate the limitations of current in-context RL methods on complex environments and discuss future directions. To facilitate future research, we release datasets for four of the considered environments.

## 1 INTRODUCTION

In-context Learning (ICL) is the ability of a model to learn new tasks by leveraging a few exemplars in its context [Brown et al., 2020]. Large Language Models (LLMs) exhibit this capability after pre-training on large amounts of data crawled from the web. A similar trend has emerged in the field of RL, where agents are pre-trained on datasets with an increasing number of tasks [Chen et al., 2021; Janner et al., 2021; Reed et al., 2022; Lee et al., 2022; Brohan et al., 2022; 2023]. After training, such an agent is capable of learning new tasks by observing previous trials in its context [Laskin et al., 2022; Liu & Abbeel, 2023; Lee et al., 2023; Raparthy et al., 2023]. Consequently, ICL is a promising direction for generalist agents to acquire new tasks without the need for re-training, fine-tuning, or providing expert-demonstrations.

Existing methods for in-context RL rely on keeping entire episodes in their context [Laskin et al., 2022; Lee et al., 2023; Kirsch et al., 2023; Raparthy et al., 2023]. Consequently, these methods face challenges in complex environments, as complex environments are usually characterized by long episodes and sparse rewards. Episodes in RL may consist of thousands of interaction steps, and processing them is computationally expensive, especially for network architectures such as the Transformer [Vaswani et al., 2017]. Furthermore, not all information an agent encountered in the past may be necessary to solve the new task. Therefore, we address the question of how to facilitate ICL for environments with long episodes and sparse rewards.

We introduce Retrieval-Augmented Decision Transformer (RA-DT), which incorporates an external memory into the Decision Transformer [Chen et al., 2021, DT] architecture (see Figure 1). Our external memory enables efficient storage and retrieval of past experiences, that are relevant for the current situation. We achieve this by leveraging a vector index populated with sub-trajectories, in combination with maximum inner product search; akin to Retrieval-augmented Generation (RAG) in LLMs [Khandelwal et al., 2019; Lewis et al., 2020; Borgeaud et al., 2022]. To encode retrieved sub-trajectories, RA-DT relies on a pre-trained embedding model, which can either be domain-specific, such as a DT trained on the same domain, or a domain-agnostic language model (LM) (see Section 3). Subsequently, RA-DT uses cross-attention to leverage the retrieved sub-trajectories and predict

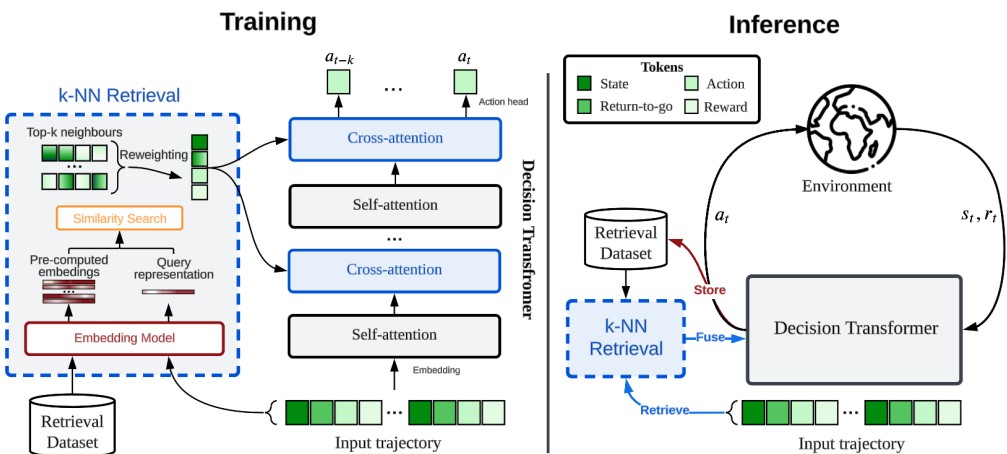

**Figure 1:** Illustration of **Retrieval-augmented Decision Transformer (RA-DT)**. **Left:** Prior to training, we encode pre-collected trajectories via an embedding model. During training, we retrieve sub-trajectories using the current context as query, and fuse them into layers via cross-attention. **Right:** During inference, the collected experience is stored and retrieved during environment interaction.

the next action. This way, RA-DT does not rely on a long context and can deal with sparse reward settings.

We evaluate the effectiveness of RA-DT on grid-world environments used in prior work with sparse rewards and increasing grid-sizes (Dark-Room, Dark Key-Door, Maze-Runner), robotics environments (Meta-World, DMControl) and procedurally-generated video games (Procgen). On grid-worlds, RA-DT considerably outperforms previous in-context RL methods, while only using a fraction of their context length. Further, we show that our domain-agnostic trajectory embedding model reaches performance close to a domain-specific one. On the remaining more complex environments, we observe consistent improvements for RA-DT on hold-out tasks, but no in-context improvement for any method. Therefore, we discuss the current limitations of RA-DT and other in-context RL methods and elaborate on potential remedies and future directions for in-context RL.

We make the following **contributions**:

- We introduce Retrieval-augmented Decision Transformers (RA-DT) and evaluate its effectiveness on a number of diverse domains.
- We show that a domain-agnostic embedding model can be utilized for retrieval in RL without requiring any pre-training, and reaches performance close to a domain-specific model.
- We release datasets for Dark-Room, Dark Key-Door, Maze-Runner, and Procgen to foster future research on in-context decision-making that leverages offline pre-training.

## 2 RELATED WORK

**In-context Learning.** ICL is a form of Meta-learning, also referred to as learning-to-learn [Schmidhuber, 1987]. Typically, meta-learning is *targeted* and learned through a meta-training phase, for example in supervised-learning [Santoro et al., 2016; Mishra et al., 2018; Finn et al., 2017] or in RL [Wang et al., 2016; Duan et al., 2016; Kirsch et al., 2019; Flennerhag et al., 2019]. In contrast, ICL *emerges* as a result of pre-training on a certain data distribution [Chan et al., 2022]. This ability was first observed in Hochreiter et al. [2001] via LSTMs [Hochreiter & Schmidhuber, 1997] and later re-discovered in LLMs [Brown et al., 2020]. Ortega et al. [2019] found that every memory-based architecture may exhibit such capabilities. Another crucial factor is a training distribution comprising a vast amount of tasks [Chan et al., 2022; Kirsch et al., 2022]. Recent works combined these properties to induce ICL in RL [Laskin et al., 2022; Lee et al., 2022; Kirsch et al., 2023]. While promising, they require keeping entire episodes in context, which is difficult in environments with long episodes. Raparthy et al. [2023] consider an in-context imitation learning setting given expert demonstrations. In contrast, RA-DT can handle long episodes and does not rely on expert demonstrations.

**Retrieval-augmented Generation.** The aim of retrieval-augmentation is to provide a model access to an external memory. This alleviates the need to store the training data in the parameters of a model and allows to condition on new data without re-training. RAG is successfully applied in the realm of LLMs [Khandelwal et al., 2019; Guu et al., 2020; Lewis et al., 2020; Borgeaud et al., 2022; Izacard et al., 2022; Ram et al., 2023], multi-modal language generation [Hu et al., 2023; Yasunaga et al., 2023; Yang et al., 2023b; Ramos et al., 2022], and for chemical reaction prediction [Seidl et al., 2022]. In RL, the access to an external memory is often referred to as episodic memory [Sprechmann et al., 2018; Blundell et al., 2016; Pritzel et al., 2017]. Goyal et al. [2022] investigate the effect of different data sources in the external memory of an online RL agent. [Humphreys et al., 2022] provide access to millions of expert demonstrations via RAG in the game of Go. In contrast, RA-DT does not rely on expert demonstrations, but leverages RAG to learn new tasks entirely in-context without the need for weight updates. Further, RA-DT does *not* rely on a pre-trained domain-specific embedding model, as we demonstrate that the embedding model can be entirely domain-agnostic.

**External memory in RL.** Most prior works have explored the utility of an external memory to cope with partially observable environments [Åström, 1965; Kaelbling et al., 1998], in which the agent must remember past events to approximate the true state of the environment. This is difficult, especially for complex tasks with sparse rewards [Arjona-Medina et al., 2019; Patil et al., 2022; Widrich et al., 2021] and long episodes. To cope with this problem, Neural Turing Machines [Graves et al., 2014], which rely on a neural controller to read from and write to an external memory, were applied to RL [Zaremba & Sutskever, 2015]. Memory networks [Weston et al., 2015] leverage an external memory for reasoning. Wayne et al. [2018] propose a memory architecture with read/write access to learn what information to store based on a world model. In contrast, RA-DT only retrieves pieces of past information similar to the current encountered situation. Hill et al. [2021] propose an attention-based external memory, where queries, keys, and values are represented by different modalities. Similarly, our domain-agnostic embedding model extends the idea of history compression via LLMs [Paischer et al., 2022; 2023] to retrieval, where queries and keys are encoded in the language space, while values comprise raw sub-trajectories.

## 3 METHOD

### 3.1 BACKGROUND

**Reinforcement Learning.** We formulate our problem setting as a Markov Decision Process (MDP) that is represented by a 4-tuple of $(\mathcal{S}, \mathcal{A}, \mathcal{P}, \mathcal{R})$. $\mathcal{S}$ and $\mathcal{A}$ denote state and action spaces, respectively. At timestep $t$ the agent observes state $s_t \in \mathcal{S}$ and issues action $a_t \in \mathcal{A}$. For each executed action, the agent receives a scalar reward $r_t$, which is given by the reward function $\mathcal{R}(r_t \mid s_t, a_t)$. $\mathcal{P}(s_{t+1} \mid s_t, a_t)$ constitutes a probability distribution over next states $s_{t+1}$ when issuing action $a_t$ in state $s_t$. RL aims at learning a policy $\pi(a_t \mid s_t)$ that predicts action $a_t$ in state $s_t$ that maximizes $r_t$.

**Decision Transformer.** Decision Transformer [Chen et al., 2021, DT] learns a policy from offline data by conditioning on future rewards. This allows rephrasing RL as a sequence modelling problem, where the agent is trained in a supervised manner to map future rewards to actions, often referred to as upside-down RL [Schmidhuber, 2019]. To train the DT, we assume access to a pre-collected dataset $\mathcal{D} = \{\tau_i \mid 1 \le i \le N\}$ of $N$ trajectories $\tau_i$ that are sampled from the environment via a behavioural policy $\pi_\beta$. Each trajectory $\tau \in \mathcal{D}$ consists of state, action, reward, and return-to-go (RTG) quadruplets $\tau_i = (s_0, a_0, r_0, \hat{R}_0, \ldots, s_T, a_T, r_T, \hat{R}_T)$, where $T$ represents the length of trajectory $\tau_i$, and $\hat{R}_t = \sum_{t'=t}^{T} r_{t'}$. The DT $\pi_\theta$ is trained to predict the ground truth action $a_t$ conditioned on sub-trajectories via cross-entropy or mean-squared error loss, depending on the domain:

$$a_t \sim \pi_\theta(a_t \mid s_{t-C:t}, \hat{R}_{t-C:t}, a_{t-C:t-1}, r_{t-C:t-1}), \tag{1}$$

where $C \le T$ is the context length. During inference, the DT is conditioned on a high RTG to produce a likely sequence of actions that yields high reward behaviour.

### 3.2 RETRIEVAL-AUGMENTED DECISION TRANSFORMER (RA-DT)

Processing long sequences with DTs is computationally expensive due to the quadratic complexity of the Transformer architecture. To address this challenge, we introduce RA-DT, which equips the DT

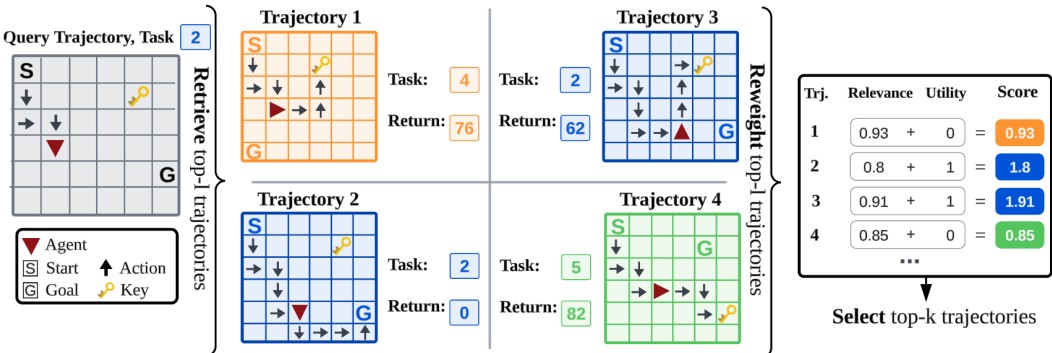

**Figure 2:** Illustration of **experience reweighting**. Given a query trajectory, we retrieve the top $l > k$ most *relevant* experiences by maximum inner product search. Each experience has an associated task ID, and return, based on which we compute their *utility*. We reweight by $s_{rel}$ and $s_u$, to obtain the final retrieval score $s_{ret}$, and return the top-$k$ experiences.

with an external memory that relies on a vector index for retrieval. Consequently, RA-DT consists of a parametric and a non-parametric component, reminiscent of complementary learning systems [Mcclelland et al., 1995; Kumaran et al., 2016]. The former is represented by the DT and learns to predict actions conditioned on the future return. The latter is the retrieval component that searches for relevant experiences, similar to Borgeaud et al. [2022] (see Figure 1).

### 3.2.1 VECTOR INDEX FOR RETRIEVAL AUGMENTATION

We aim at augmenting the DT with a vector index (external memory) that allows for retrieval of relevant experiences. To this end, we build our vector index by leveraging an embedding model $g : \tau \mapsto \mathbb{R}^{d_r}$ that takes a trajectory $\tau$ and returns a vector of size $d_r$. Given a dataset $\mathcal{D}$ of trajectories, we obtain a set of key-value pairs of our vector index by embedding all sub-trajectories $\tau_{t-C:t} \in \mathcal{D}$ via $g(\cdot)$ to obtain $\mathcal{K} \times \mathcal{V} = \{(g(\tau_{i,t-C:t}), \tau_{i,t-C:t+C}) \mid 1 \leq i \leq |\mathcal{D}|\}$. Note that values contain sub-trajectories ranging from $t - C$ to $t + C$, while keys use sub-trajectories $t - C : t$ for a fixed $C$, where $t$ goes over trajectory length in increments of $C$ (see Appendix C.4 for more details). The reason for this choice is that during inference, the model does not have access to future states.

In RAG applications for Natural Language Processing (NLP), a common choice for $g(\cdot)$ is a pre-trained LM. While pre-trained models in NLP are ubiquitous, they are rarely available in RL. A natural choice to instantiate $g(\cdot)$ is to train a DT on the pre-collected dataset $\mathcal{D}$, as they exhibit a well-separated embedding space after pre-training [Schmied et al., 2024]. Therefore, they are well suited for retrieval since a new task can be matched to similar tasks in the vector index. As a domain-agnostic alternative, we propose to utilize the FrozenHopfield (FH) mechanism Paischer et al. [2022] to map trajectories to the embedding space of a pre-trained LM. This enables instantiating $g(\cdot)$ with a pre-trained language encoder. The FH mechanism is parameterized by an embedding matrix $\boldsymbol{E} \in \mathbb{R}^{v \times d_{\mathrm{LM}}}$ of a pretrained LM with vocabulary size $v$ and hidden dimension $d_{\mathrm{LM}}$, a random matrix $\boldsymbol{P}$ with entries sampled from $\mathcal{N}(0, d_{\mathrm{in}}/d_{\mathrm{LM}})$, and a scaling factor $\beta$ and performs:

$$\mathrm{FH}(\boldsymbol{x}_t) = \boldsymbol{E}^\top \mathrm{softmax}(\beta \boldsymbol{E} \boldsymbol{P} \boldsymbol{x}_t). \tag{2}$$

We denote $\boldsymbol{x}_t \in \mathbb{R}^{d_{\mathrm{in}}}$ as the input token and apply the FH position-wise to every state/action/reward token in a sub-trajectory $\tau_{t-C:t}$ separately. Finally, we apply a LM on top of the FH to obtain the keys of our vector index by setting $g(\cdot) = \mathrm{LM}(\mathrm{FH}(\cdot))$. Utilizing the FH enables leveraging the expressive power of pre-trained LMs as trajectory encoders for RL. This sidesteps the need for pre-training a domain-specific model and can be incorporated in any existing retrieval-augmentation pipeline.

### 3.2.2 SEARCHING FOR SIMILAR EXPERIENCES

Given an input sub-trajectory $\tau_{\mathrm{in}} \in \mathcal{D}$, we first construct a query $\boldsymbol{q} = g(\tau_{\mathrm{in}})$, using our embedding model $g(\cdot)$ (see Appendix C.4 for details). Then, we use maximum inner product search (MIPS)

between $q$ and all keys $k \in \mathcal{K}$ and select the corresponding top-$l$ sub-trajectories $\tau_{\text{ret}} \in \mathcal{V}$ by:

$$\mathcal{R} = \arg \max_{k \in \mathcal{K}}^{l} \text{cossim}(q, k), \tag{3}$$

where $\text{cossim}(q, k) = \frac{q \cdot k}{\|q\| \|k\|}$ is the cosine similarity. Consequently, $\mathcal{R}$ contains the set of retrieved sub-trajectories and their keys. Providing too similar experiences to the model may hinder learning [Yasunaga et al., 2023] and we apply retrieval regularization during training (see Appendix C.4).

### 3.2.3 REWEIGHTING RETRIEVED EXPERIENCES

Following Park et al. [2023], we characterize the usefulness of retrieved sub-trajectories in $\mathcal{R}$ along two dimensions: *relevance* and *utility*. The relevance of a key $k \in \mathcal{K}$ is defined by its cosine similarity to the query $q$. While a retrieved experience may be relevant, it might not be important. Determining the utility of a sequence in general is hard. Thus, we experiment with two heuristics that follow different definitions of utility. The first assigns more utility to sub-trajectories with high return, and is utilized *at inference* only. The second assigns utility to sub-trajectories that originate from the same task as the query and is used *at training* only. Then, we reweight a retrieved experience according to:

$$s_{\text{ret}}(k, q, \tau_{\text{ret}}) = s_{\text{rel}}(k, q) + \alpha \, s_{\text{u}}(\tau_{\text{ret}}, \tau_{\text{in}}), \tag{4}$$

where $s_{rel} = \text{cossim}(k, q)$ and $s_{\text{u}}$ measures the utility of a retrieved sub-trajectory weighted by $\alpha$. Note that we instantiate $s_{\text{u}}(\cdot, \cdot)$ differently depending on whether the agent is in training or inference mode. At *training* time, a pre-collected set of trajectories that contains multiple tasks is stored in the vector index (Figure 1, left). Trajectories can be obtained from human demonstrations or RL agents. Therefore, we encourage the agent to retrieve sub-trajectories of the same task. During training, we use: $s_{\text{u}}(\tau_{\text{ret}}, \tau_{\text{in}}) = \mathbb{1}(\text{t}(\tau_{\text{ret}}) = \text{t}(\tau_{\text{in}}))$, where $\text{t}(\cdot)$ takes a sub-trajectory and returns its task index.

During *inference*, we evaluate the ICL capabilities of the agent. Starting from an *empty* vector index, we store experiences of the agent while it interacts with the environment (see Figure 1, right). Thus, during inference, the agent can only retrieve experiences from the same task. Therefore, we steer the agent to produce high reward behaviour on the new task by reweighting a retrieved sub-trajectory by the total return achieved over the episode it appears in, i.e., $s_{\text{u}}(\tau_{\text{ret}}, \tau_{\text{in}}) = \sum_{i=0}^{T} r_i$. We apply this reweighting to the retrieved experiences in $\mathcal{R}$ and select the top-$k$ elements by:

$$\mathcal{S} = \arg \max_{k, \tau_{\text{ret}} \in \mathcal{R}}^{k} s_{\text{ret}}(k, q, \tau_{\text{ret}}), \tag{5}$$

where we normalize both scores to be in the range $[0, 1]$, such that they contribute equally to the final weight. Our reweighting mechanism is illustrated in Figure 2.

### 3.2.4 INCORPORATING RETRIEVED EXPERIENCES

After reweighting, the set $\mathcal{S}$ contains sub-trajectories that are both important and relevant for the current input $\tau_{\text{in}}$ to the DT $\pi_\theta$. To incorporate the retrieved experiences in the DT, we interleave it with cross-attention layers (CA) after every self-attention (SA) layer. The retrieved sub-trajectories are encoded by separate embedding layers for each token type (state/action/reward/RTG) and then passed to the CA layers. Thus, our RA-DT predicts actions $a_t$ given input trajectory and retrieved trajectory by:

---

**Algorithm 1** In-context Learning with RA-DT

**Input:** DT $\pi_\theta$, embed model $g$, episodes $N$, episode len $T$, context len $C$, `retrieve`, `reweight`.
1: $\mathcal{I} \leftarrow \emptyset$            ▷ Inititalize index
2: **for** $1 \ldots N$ **do**
3:    $s, \tau \leftarrow \text{env.reset}(), \emptyset$
4:    **for** $t = 1 \ldots T$ **do**
5:       $q = g(\tau_{t-C:t})$    ▷ Construct query
6:       $\mathcal{R} \leftarrow \text{retrieve}(q, \mathcal{I})$ ▷ Top-$l$ trjs, Eq. 3
7:       $\mathcal{S} \leftarrow \text{reweight}(\mathcal{R})$   ▷ Top-$k$, Eq. 4, 5
8:       $a \sim \pi_\theta(a \mid \tau_{t-C:t}, \{\tau_{\text{ret}} \in \mathcal{S}\})$   ▷ Predict
9:       $s', r \leftarrow \text{env.step}(a)$
10:      $\tau \leftarrow \tau \cup (s, a, r)$ ▷ Append transition to $\tau$
11:      $s \leftarrow s'$
12:    **end for**
13:    $\mathcal{I} \leftarrow \mathcal{I} \cup \tau$    ▷ Add trajectory $\tau$ to index $\mathcal{I}$
14: **end for**

---

$$a_t \sim \pi_\theta(a_t \mid \tau_{\text{in}}, \{\tau_{\text{ret}} \in \mathcal{S}\}). \tag{6}$$

In Algorithm 1, we show the pseudocode for in-context RL with RA-DT at *inference* time. In addition, we show RA-DT at *training* time in Algorithm 2 of Appendix C.4.

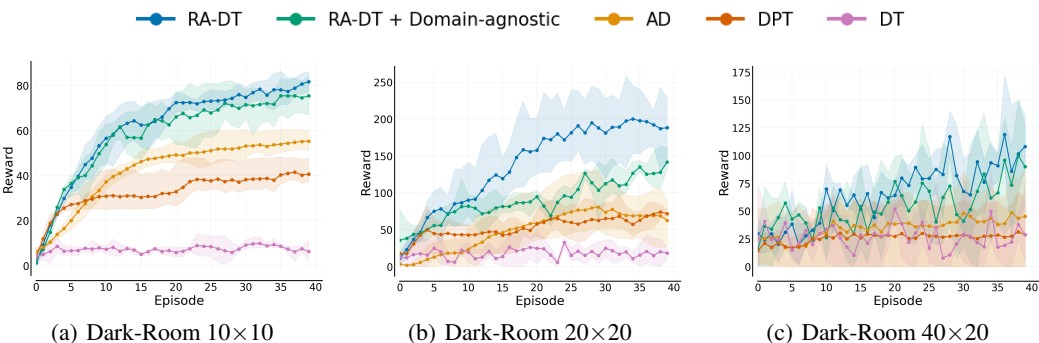

(a) Dark-Room 10×10      (b) Dark-Room 20×20      (c) Dark-Room 40×20

**Figure 3:** ICL performance on **Dark-Room (a)** 10×10, **(b)** 20×20, **(c)** 40×20 at end of training (100K steps). We evaluate each agent for 40 episodes on each of the 20 **evaluation tasks** and report mean reward (+ 95% CI) over 3 seeds.

## 4 EXPERIMENTS

We evaluate the ICL abilities of RA-DT on grid-world environments used in prior works, namely Dark-Room (see Section 4.1), Dark Key-Door (Section 4.2), and MazeRunner (Section 4.3) [Laskin et al., 2022; Lee et al., 2022; Grigsby et al., 2023], with increasingly larger grid-sizes, resulting in longer episodes. Moreover, we evaluate RA-DT on two robotic benchmarks (Meta-World and DMControl, Section 4.4) and procedurally-generated video games (Procgen, Section 4.5).

Across experiments, we report performances for two variants of **RA-DT**. The first variant leverages a domain-specific embedding model for retrieval, specifically a DT trained on the same domain. The second variant (**RA-DT + Domain-agnostic**) makes use of the FH mechanism in combination with BERT [Devlin et al., 2019] as the pre-trained LM. Consequently, this variant of RA-DT does not require any domain-specific pre-training of the embedding model. We compare RA-DT against the vanilla **DT** and two established in-context RL methods, namely Algorithm Distillation [Laskin et al., 2022, **AD**] and Decision Pre-trained Transformer [Lee et al., 2023, **DPT**]. Following, Agarwal et al. [2021] we report the mean across tasks and 95% confidence intervals over 3 seeds. We use a context length equivalent to two episodes (from 200 up to 2000 timesteps) for AD, DPT and DT. For RA-DT, we use a considerably shorter context length of 50 transitions, unless mentioned otherwise. On grid-worlds, we train all methods for 100K steps and evaluate after every 25K steps. Similarly, we train for 200K steps and evaluate after every 50K steps for Meta-World, DMControl and Procgen. All grid-worlds and Procgen exhibit discrete actions and consequently, we train all methods via the cross-entropy loss to predict the next actions. On Meta-World and DMControl, we train all method using the mean-squared error loss to predict continuous actions. Following Laskin et al. [2022] and Lee et al. [2023], our primary evaluation criterion is performance improvement during ICL trials. After training, the agent interacts with the environment for a fixed amount of episodes, each of which is considered a single trial. Upon completion of an ICL trial, the respective episode is stored in the vector index. We provide further training and implementation details in Appendix C.

### 4.1 DARK-ROOM

**Experiment Setup.** Dark-Room is commonly used in prior work on in-context RL [Laskin et al., 2022; Lee et al., 2023]. The agent is located in an empty room, observes only its x-y coordinates, and has to navigate to an invisible goal state ($|\mathcal{S}| = 2$, $|\mathcal{A}| = 5$, see Figure 9). A reward of +1 is obtained in every step the agent is located in the goal state. Because of partial observability, it must leverage memory of previous episodes to find the goal. We conduct experiments on three different grid sizes, namely 10×10, 20×20, and 40×20, and corresponding episode lengths of 100, 200 and 800, respectively. We designate 80 and 20 randomly assigned goals as train and evaluation locations, respectively, as in Lee et al. [2023]. We use Proximal Policy Optimization (PPO) [Schulman et al., 2017] to generate 100K transitions per goal for 10×10 and 20×20 grids and 200K for 40×20 (see Figure 7 for single task expert scores). During evaluation, the agent interacts with the environment

for 40 ICL trials, and we report the scores at the last evaluation step (100K). We provide additional details on the environment, the generated data, and the training procedure in Appendix B.1 and C.

**Results.** In Figure 3, we show the ICL performances on the 20 hold-out tasks for all considered methods on Dark-Room **(a)** 10×10, **(b)** 20×20, and **(c)** 40×20. In addition, we present the ICL curves on the training tasks and the learning curves across the entire training period in Figures 14 and 15 in Appendix D.1. Overall, we observe that RA-DT attains the highest average rewards on all 3 grid-sizes at the end of the 40 ICL-trials. On 10×10, RA-DT obtains near-optimal performance scores both with the domain-specific and domain-agnostic embedding model. The vanilla DT does not exhibit any performance improvement across trials. This indicates the improvement in performance for RA-DT can be attributed to the retrieval component. Furthermore, RA-DT outperforms AD and DPT without keeping entire episodes in its context window. Similarly, RA-DT outperforms all baselines on the 20×20 and 40×20 grids. While RA-DT successfully improves in-context, the baselines exhibit only little learning progress over the ICL trials, especially for larger grid sizes. However, the final performance scores for 20×20 and 40×20 are not optimal. With increasing grid size, discovering the goal requires systematic exploration in combination with targeted exploitation. Therefore, we conduct a qualitative analysis on the exploration behaviour of RA-DT. We find that RA-DT develops strategies to imitate a given successful context (see Figure 16), and avoids low-reward routes given an unsuccessful context (see Figure 17).

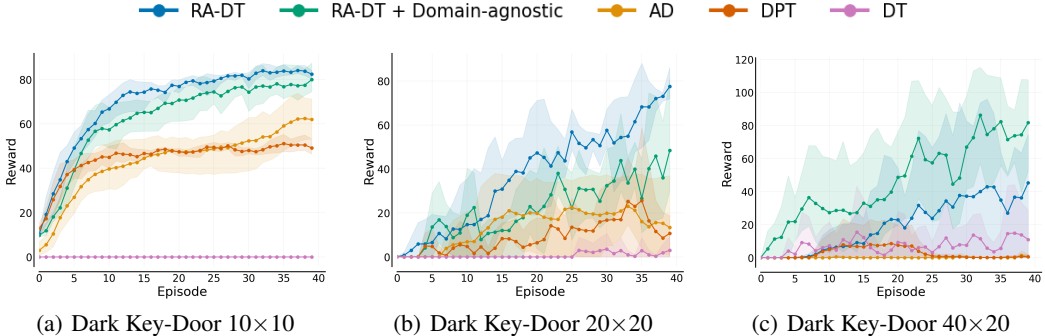

(a) Dark Key-Door 10×10     (b) Dark Key-Door 20×20     (c) Dark Key-Door 40×20

**Figure 4:** ICL performance on **Dark Key-Door (a)** 10×10, **(b)** 20×20, **(c)** 40×20 at end of training (100K steps). We evaluate each agent for 40 episodes on each of the 20 **evaluation tasks** and report mean reward (+ 95% CI) over 3 seeds.

### 4.2 DARK KEY-DOOR

**Experiment Setup.** In Dark Key-Door, the agent is located in a room with two invisible objects: a key and a door. The agent has to pick up the invisible key, then navigate to the door. Because of the presence of two key events, the task-space is combinatorial in the number of grid-cells ($100^2 = 10000$ possible tasks for $10 \times 10$) and is therefore considered more difficult. A reward of +1 is obtained once for picking up the key and for every step the agent stands on the door grid-cell after it collected the key. We retain the same experiment setup as in Section 4.1 and provide further details in Appendix B.1 (also see Figure 8 for single-task expert scores).

**Results.** On $10 \times 10$ and $20 \times 20$, RA-DT outperforms baselines, with the performance ranking remaining the same as on Dark-Room (see Figure 4). Surprisingly, domain-agnostic RA-DT outperforms its domain-specific counterpart on $40 \times 20$, which demonstrates that the domain-agnostic embedding model is a promising alternative. This result indicates that RA-DT can successfully handle environments with more than one key event, even with shorter observed context.

### 4.3 MAZE-RUNNER

**Experiment Setup.** Maze-Runner was introduced by Grigsby et al. [2023] and inspired by Pasukonis et al. [2022]. The agent is located in a procedurally-generated $15 \times 15$ maze (see Figure 10), observes continuous Lidar-like depth representations of states, and has to navigate to one, two, or three goal locations in the correct order ($|\mathcal{S}| = 6, |\mathcal{A}| = 4$). A reward of +1 is obtained when reaching a goal

location. Episodes last for a maximum of 400 steps, or terminate early if all goal locations have been visited. Similar to Dark-Room, we use PPO to generate 100K environment interactions for 100 procedurally-generated mazes. We train all methods on a multi-task dataset that comprises trajectories from 100 mazes, evaluate on 20 unseen mazes, and report performance over 30 ICL trials. We give further details on the environment, the dataset, and the experiment setup in Appendix B.2 and D.2.

**Results.** We find that RA-DT considerably outperforms all baselines in terms of final performance (see Figure 5). Surprisingly, RA-DT is the only method to improve over the course of the 30 ICL trials. However, we observe a considerable performance gap between train mazes and test mazes (0.65 vs. 0.4 reward, see Figure 20), indicating that solving unseen mazes requires an enhanced ability to generalize and learn from previous trials.

## 4.4 META-WORLD & DMCONTROL

**Experiment Setup.** Next, we evaluate RA-DT on two multi-task robotics benchmarks, Meta-World [Yu et al., 2020b] and DMControl [Tassa et al., 2018]. States and actions in both benchmarks are multidimensional continuous vectors. While the state and action space in Meta-World remain constant across all tasks ($|\mathcal{S}| = 39$, $|\mathcal{A}| = 6$), they vary considerably in DM-Control ($3 \leq |\mathcal{S}| \leq 24$, $1 \leq |\mathcal{A}| \leq 6$). Episodes last for 200 and 1000 steps in Meta-World and DMControl, respectively. We leverage the datasets released by Schmied et al. [2024]. For Meta-World, we pre-train a multi-task policy on 45 of the 50 tasks (ML45, 90M transitions in total) and evaluate on the 5 remaining tasks (ML5). Similarly, on DMControl, we pre-train on 11 tasks (DMC11, 11M transitions in total) and evaluate on 5 unseen tasks (DMC5). We provide further details on the environments, datasets, and experiment setup in Appendices B.3 and D.3, and B.4 and D.4 for Meta-World and DMControl, respectively.

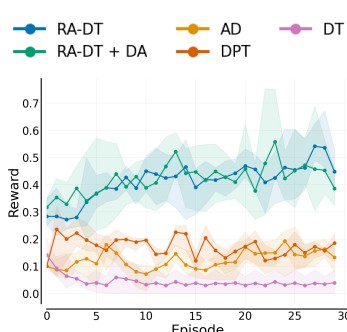

**Figure 5:** ICL on **MazeRunner**. We evaluate over 30 ICL trials and report the mean reward (+ 95% CI) over 3 seeds.

**Results.** We present the learning curves and corresponding ICL curves for Meta-World and DM-Control in Figure 22 and 23, and Figures 24 and 25 in Appendix D, respectively. In addition, we provide the raw and data-normalized scores in Tables 3 and 4, respectively. On both benchmarks, we find that RA-DT attains considerably higher scores on unseen evaluation tasks, but slightly lower average scores across training tasks compared to DT. However, these performance gains on evaluation tasks are not reflected in improved ICL performance. In fact, we only observe slight in-context improvement on training tasks, but not on holdout tasks for any of the considered methods.

## 4.5 PROCGEN

**Experiment Setup.** Finally, we conduct experiments on Procgen [Cobbe et al., 2020], a benchmark consisting of 16 procedurally-generated video games, designed to test the generalization abilities of RL agents. The procedural generation in Procgen is controlled by setting an environment seed, which results in visually diverse observations for the same underlying task (see `starpilot`-example in Figure 12). In Procgen, the agent receives image-based inputs ($|\mathcal{S}| = 3 \times 64 \times 64$). All 16 tasks share a discrete action space ($|\mathcal{A}| = 15$). Rewards are either dense or sparse depending on the environment.

We follow Raparthy et al. [2023] and use 12 tasks for training (PG12) and 4 tasks for evaluation (PG4). First, we generate datasets by training task-specific PPO agents for 25M timesteps on 200 environment seeds per task in `easy` difficulty. Then, we pre-train a multi-task policy on the PG12 datasets (24M transitions in total, 2M per task). We leverage the procedural generation of Procgen and evaluate all models in three settings: *training tasks - seen* (PG12-Seen), *training tasks - unseen* (PG12-Unseen), and *evaluation tasks - unseen* (PG4). Additional details on the generated datasets and our environment setup are available in Appendices B.5 and D.5.

**Results.** Similar to our results on Meta-World and DMControl, we find that RA-DT improves average performance scores across all three settings compared to the baselines (see Figure 26 and Tables 5, 6,

7 in Appendix D.5), but no method exhibits in-context improvement during evaluation (Figure 27). We further discuss our negative results on Procgen, Meta-World, and DMControl in Section 5.

### 4.6 ABLATIONS

To better understand the effect of learning with retrieval, we present a number of ablation studies on essential components in RA-DT conducted on Dark-Room $10 \times 10$ (more details in Appendix E).

**Retrieval outperforms sampling of experiences.** To investigate the effect of learning with retrieved context, we substitute retrieval with random sampling, either over all tasks, or from the same task (see Figure 6a). We find that training with retrieval outperforms both sampling variants, highlighting the benefit of training with retrieval to improve ICL abilities. We hypothesise this is because retrieval constructs bursty sequences, which was found to be important for ICL [Chan et al., 2022].

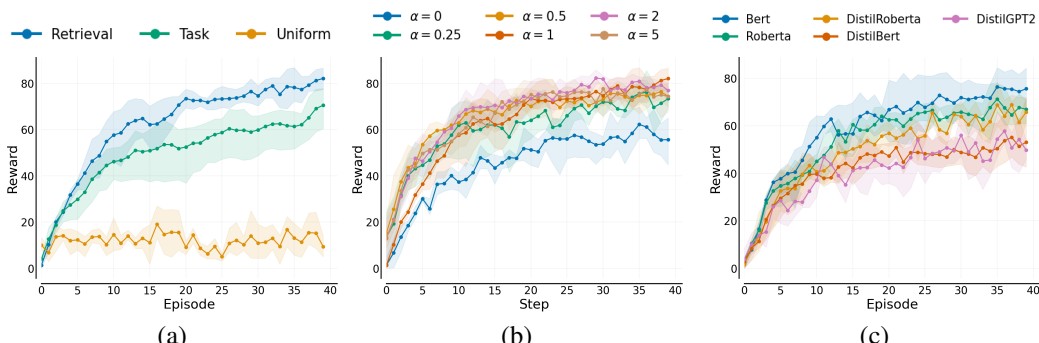

(a)             (b)             (c)

**Figure 6:** Ablations on important components in RA-DT conducted on Dark-Room $10 \times 10$. We show **(a)** the effect of training with retrieval vs. sampling, **(b)** a sensitivity analysis on $\alpha$ as used in the re-weighting mechanism during training, and **(c)** the effect of leveraging different LMs as pre-trained embedding models for domain-agnostic retrieval.

**Reweighting Experiences.** RA-DT reweights a sub-trajectory by its *relevance* and *utility* score. By default, we use task-based reweighting during training. In Figure 28, we compare against alternatives, such as reweighting by return. Indeed, we find that task-based reweighting is critical for high performance, because it ensures that retrieved experiences are useful for predicting the next action.

**Sensitivity of Reweighting.** We conduct a sensitivity analysis on $\alpha$ used in the reweighting mechanism (see Equation 4). In Figure 6b, we find that RA-DT performs well for a range of values for $\alpha$ used during training, but performance declines if no re-weighting is employed ($\alpha = 0$). We perform the same analysis for $\alpha$ during evaluation in Figure 29.

**Effect of Retrieval Regularization.** We evaluate with three retrieval regularization strategies to mitigate the effect of copying the context: deduplication, similarity cut-off, and query dropout. To evaluate their impact on ICL performance, we systematically removed each one from RA-DT (see Figure 30). We found the combination of all three to be effective and add them to our pipeline.

**Different LMs for domain-agnostic RA-DT.** Finally, we investigate how strongly domain-agnostic RA-DT is influenced by the choice of pre-trained LM for the embedding model. We compare our default choice BERT against other smaller/larger LMs (see Figure 36). We found that BERT performs best and performance decreases with smaller models.

**Effect of Retrieval on Training/Inference Efficiency.** Retrieval-augmentation adds computational overhead to the training/inference pipeline due to the cost of embedding the query and searching for similar experiences. However, we find that RA-DT results in significantly faster training times because of shorter context length (up to $7\times$ see Appendix E.7). At inference-time, RA-DT is slightly slower compared to baselines when retrieving at every step, but exhibits similar inference speeds when retrieving less frequently (see Appendix E.8). Importantly, the retrieval mechanism in RA-DT enables access to the entirety of the experiences collected across all ICL trials with small additional cost. The ability to access a broader range of experiences may be a reason for its enhanced performance.

For additional ablations on RA-DT and on our baselines, we refer to Appendix E.

## 5 DISCUSSION

In this section, we highlight current challenges of RA-DT and other offline in-context RL methods.

**Memory-Exploitation vs. Meta-learning Abilities.** Current *offline* in-context RL methods are predominantly evaluated on contextual bandits or grid-worlds, such as Dark-Room [Laskin et al., 2022; Lee et al., 2023; Lin et al., 2023; Sinii et al., 2023; Huang et al., 2024], which can only be solved by leveraging the context. However, it remains unclear to what extent the agent learns to learn in-context or simply copies from its context. Further, in our experiments on fully-observable environments (MetaWorld, DMControl, and Procgen), we did not observe ICL behaviour (see Appendices D.3, D.4, D.5). Therefore, it is necessary that future research on in-context RL disentangles the effects of memory and meta-learning abilities, similar to memory and credit-assignment [Ni et al., 2024].

**Challenges of Next-Action Prediction.** Most in-context RL methods learn from offline datasets via next-action prediction and causal sequence modelling objectives. As such, they cannot learn to infer the utility of an action, and thus, distinguish between positive and negative examples. This can induce delusions, which lead to repetitions of suboptimal actions and copying behaviour [Ortega et al., 2021] (see Figure 19 for examples on Dark-Room). In contrast, *online* in-context RL methods have shown promising adaptation abilities [Team et al., 2023; Grigsby et al., 2023; Lu et al., 2024]. A similar trend has been observed in online meta-RL methods [Melo, 2022; Shala et al., 2024]. Consequently, a potential remedy to this problem is to train a value function to learn the utility of an action. However, this is usually not straightforward and requires constrained optimization objectives [Zanette et al., 2021; Kumar et al., 2020]. Therefore we leave this approach to future work.

**Conditioning Strategies in RL.** In LLMs, applying sophisticated conditioning strategies is important to improve ICL abilities [Wei et al., 2022; Yao et al., 2024; Agarwal et al., 2024]. Even though RTG-conditioning [Chen et al., 2021], and chain-of-hindsight [Liu & Abbeel, 2023] have shown promise for generating high reward behaviour in DTs, the broader landscape for conditioning strategies for in-context RL remains under-explored. Therefore, we believe that systematically investigating conditioning methods for in-context RL is a fruitful direction for future research.

**Diversity of the Pre-training Distribution.** The diversity and scale of the pre-training dataset may significantly affect the emergence of ICL. In our experiments, we pre-train on a relatively small set of tasks. Our results on gridworlds suggest that this is sufficient for ICL to emerge on simple environments. However, on more complex environments, the unseen tasks can be considered out-of-distribution and higher pre-training diversity may be necessary for ICL to emerge. It remains unclear how much diversity is required to elicit in-context RL, and if existing large-scale agents exhibit ICL [Reed et al., 2022; Raad et al., 2024]. One promising approach is to expand the pre-training diversity through learned interactive simulations [Yang et al., 2023a; Bruce et al., 2024].

## 6 CONCLUSION

Existing in-context RL methods keep entire episodes in their context window, which is challenging as RL environments are typically characterized by long episodes and sparse rewards. To address this challenge, we introduce RA-DT, which employs an external memory mechanism to store past experiences and to retrieve experiences relevant for the current situation. RA-DT outperforms baselines on grid-worlds, while using only a fraction of their context length. While RA-DT improves average performance on holdout tasks on complex environments, it struggles to exhibit ICL, along with other in-context RL methods. Consequently, we illuminate the current limitations of in-context RL methods and discuss future directions. Finally, we release our datasets for Dark-Room, Dark Key-Door, MazeRunner, and Procgen, to facilitate future research on in-context RL.

**Future Work.** Besides the general directions discussed in Section 5, we highlight a number of concrete approaches to extend RA-DT. While we focus on ICL without relying on expert demonstrations, pre-filling the external memory with demonstrations may enable RA-DT to perform more complex tasks. This may be effective for robotics applications, where expert demonstrations are easy to obtain. Furthermore, end-to-end training of the retrieval component in RA-DT, similar to [Izacard et al., 2022], may result in more precise context retrieval and enhanced down-stream performance. Finally, we envision that modern recurrent architectures [Bulatov et al., 2022; Gu & Dao, 2023; Beck et al., 2024] as policy backbones may benefit RA-DT by maintaining hidden states across many episodes.

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

APPENDIX

# Contents

## A   ETHICS STATEMENT & REPRODUCIBILITY

In recent years, there has been a trend in RL towards large-scale multi-task models that leverage offline pre-training. In this work, we broadly aim at building agents that can learn new tasks via ICL without the need for re-training or fine-tuning. Our goal is to reduce the need to provide entire past episodes in the agent's context, by augmenting the agent with an external memory in combination with a retrieval component, similar to RAG in LLMs. We believe that multi-task agents of the near future will be able to perform a broad range of tasks, and that these agents will greatly benefit from RAG as used in RA-DT. The external memory component can enable agents to leverage information from in its own distant past or experiences from other agents. Such agents could have an immense impact on the global economy (e.g., as a source of inexpensive labour). As such, they do not come without risks and the potential for misuse. While we believe that our work can significantly impact the positive use of future agents, it is essential to ensure responsible deployment of future technologies.

Upon publication, we will open-source the code-base used for our experiments, and release the datasets we generated. In addition, we provide further information on the environments/datasets,

implementation including hyperparameter tables, and on our experiments in Appendices B, C, D, respectively.

## B ENVIRONMENTS & DATASETS

### B.1 DARK-ROOM AND DARK KEY-DOOR

The Dark-Room environment is modelled after Morris-Watermaze, a classic experiment in behavioural neuroscience for studying spatial memory and learning in animals [D'Hooge & De Deyn, 2001]. We design our Dark-Room and Dark Key-Door environments in Minihack [Samvelyan et al., 2021], which is based on the NetHack Learning Environment [Küttler et al., 2020]. We construct grids of dimensions $10 \times 10$, $20 \times 20$ and $40 \times 20$, as depicted in Figure 9. With increasing grid sizes, the task of locating the goal becomes harder as the number of possible positions in the grid grows (100, 400, 800). Therefore, we set the number of interaction steps per environment equal to the number of grid cells. Consequently, larger grids results in longer episodes and thus context lengths (e.g., 2400 for AD). The agent observes its own x-y position on the grid and can perform one of 5 actions at every interaction step (up, down, left, right, stay). Episodes start in the top left corner (0,0) and the agent is reset to the start position after every episode.

In **Dark-Room**, the agent has to navigate to a randomly placed and invisible goal position. Therefore, the task space in Dark-Room environments is equal to the number of grid-cells (i.e., 100 for $10 \times 10$). The agent receives a reward for +1 for every step in the episode it is located in the goal position and 0 otherwise. As there are as many grid-cells as episode steps, the optimal strategy for solving the Dark-Room task is to use the first episode to visit every cell to find the hidden goal location. Once found, this knowledge can be exploited in upcoming trials.

In contrast, in **Dark Key-Door**, there are two objects: a key and a goal state. Similar to Dark-Room, the key and goal position are randomly placed on the grid. The agent has to first pick up the invisible key and then find the invisible goal. Due to the presence of the two key events (picking up the key, finding the goal), the task space is combinatorial in the number of grid-cells (i.e., $100^2 = 10000$ for $10 \times 10$). This makes the Dark Key-Door more challenging than the Dark-Room task, especially as the grid-size becomes larger.

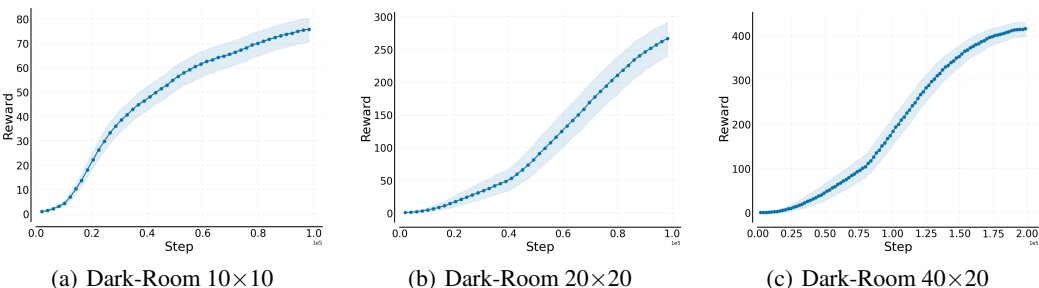

(a) Dark-Room 10×10     (b) Dark-Room 20×20     (c) Dark-Room 40×20

**Figure 7:** Average performances of the **source algorithm**, PPO, on 80 train tasks for Dark-Room **(a)** 10×10, **(b)** 20×20, and **(c)** 40×20. For (a), (b) we train PPO on individual tasks for 100K environment steps. For (c), we train for 200K environment steps to take the longer episode lengths into account. We evaluate the agents after every 10K steps. Curves show the mean reward achieved (+ 95% CI) across the 80 train tasks.

**Training Dataset.** For both Dark-Room and Dark Key-Door, we generate training datasets for 80 randomly assigned goals or key-goal combinations. We use PPO [Schulman et al., 2017] to generate 100K environment transitions per goal location for $10 \times 10$ and $20 \times 20$ grids and 200K environment transitions for the largest grid. Therefore, the total number of transitions across datasets is 8M for $10 \times 10$ and $20 \times 20$ grids and 16M for $40 \times 20$.

We train PPO with standard hyperparameter settings in `stable-baselines3` [Raffin et al., 2021] using a learning rate of $3e^{-4}$, batch size of $64$, number of steps between updates of $2048$, number of update epochs $10$ and entropy coefficient of $0.01$. For $20 \times 20$ and $40 \times 20$ grids, we increase

the number of update epochs to 30 and the entropy coefficient of to $0.1$ for $40 \times 20$. We store all generated transitions of PPO for our datasets. Consequently, the final datasets contain a mixture of suboptimal or exploratory, and optimal or exploitative behaviour.

**Source Algorithm Performance.** We show average learning curves across all task-specific PPO agents on the 80 training tasks for all grid-sizes in Figures 7 and 8 for Dark-Room and Dark Key-Door, respectively. For the $10 \times 10$ grids, the average performance converges towards optimal performance. However, on the larger grid sizes, the performances are below the optimum. This is because it takes the agent longer to discover and collect successful episodes by initially random environment interaction as the grids become larger.

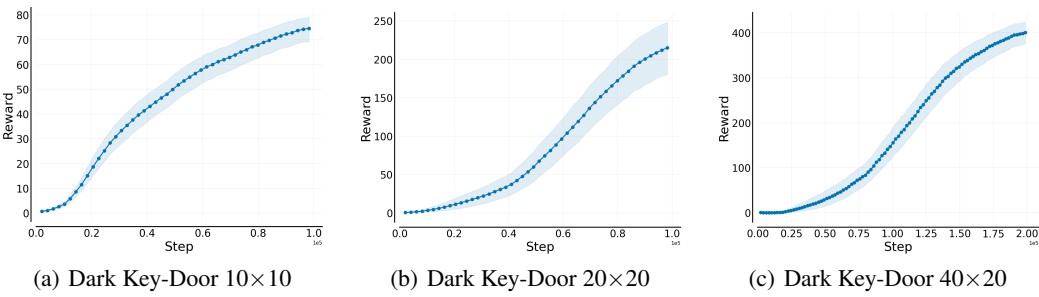

(a) Dark Key-Door 10×10   (b) Dark Key-Door 20×20   (c) Dark Key-Door 40×20

**Figure 8:** Average performances of the **source algorithm**, PPO, on 80 train tasks for **Dark Key-Door (a)** 10×10, **(b)** 20×20, and **(c)** 40×20. For (a), (b) we train PPO on individual tasks for 100K environment steps. For (c), we train for 200K environment steps. We evaluate the agents after every 10K steps. Curves show the mean reward achieved (+ 95% CI) across the 80 train tasks.

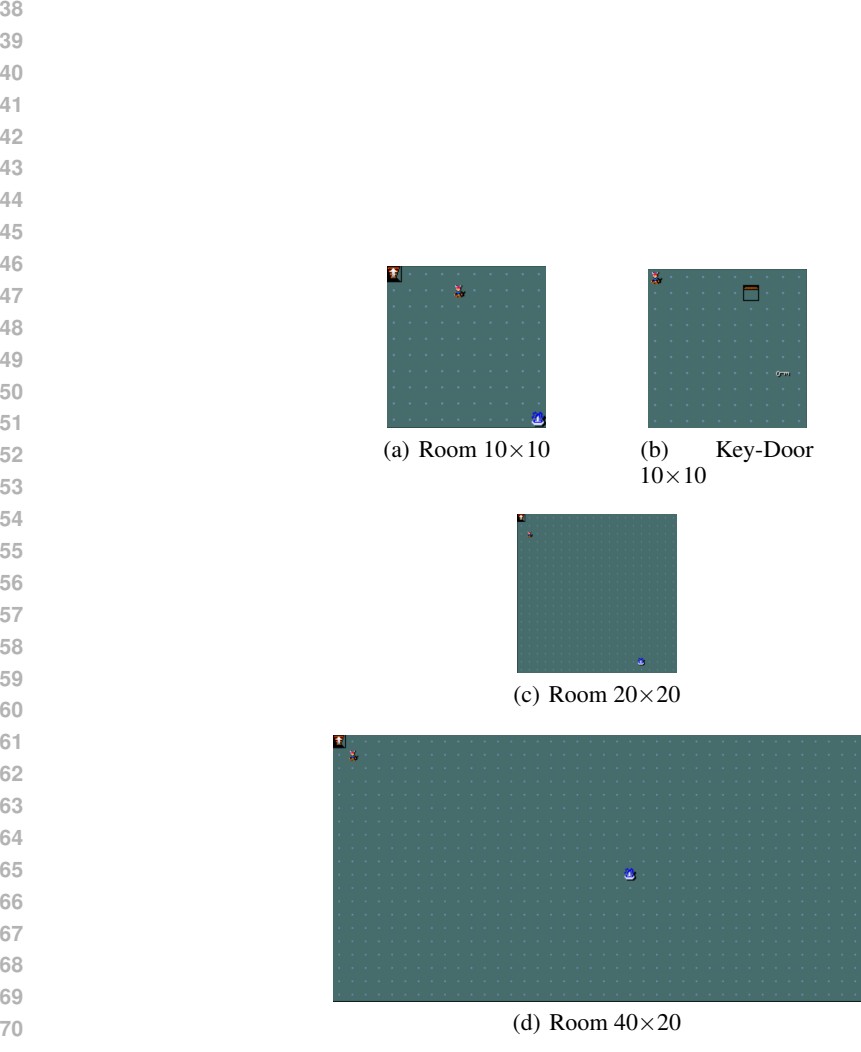

(a) Room 10×10

(b)     Key-Door
10×10

(c) Room 20×20

(d) Room 40×20

**Figure 9:** Mini-grid environments. In **Dark-Room**, the agent is located in a room and has to navigate to an invisible goal location. We use grid-sizes **(a)** 10×10, **(b)** 20×20 and **(c)** 40×20 for our experiments. In **(b) Dark-KeyDoor**, the agent has to pick up an invisible key, then navigate to the invisible goal location. Agents only observe their current x-y coordinate on the grid. Reward of +1 is obtained in every step the agent is situated in the goal state, +1 for picking up the key.

### B.2 MAZERUNNER

MazeRunner was introduced by [Grigsby et al., 2023] and inspired by the Memory Maze environment [Pasukonis et al., 2022]. The agent is located in a 15×15 procedurally-generated maze and has to navigate to a sequence of one, two, or three goal locations in the right order (see Figure 10). Similar to Dark-Room environments, MazeRunner is partially observable and exhibits sparse rewards. The agent observes a Lidar-like 6-dimensional representation of the state that contains 4 continuous values that measure the distance from the agent's location to the nearest wall, and the x-y coordinates of the agent's position in the grid. The action-space is 4-dimensional (up, down, left, right). A reward of +1 is obtained when reaching the currently active goal state in the goal sequence. Therefore, the total achievable reward is equal to the number of goal states. Episodes last for a maximum of 400 steps or terminate early, if all goal locations have been reached. After every episode, the agent (gray box in Figure 10) is reset to the origin location. During evaluation, we allow for 30 ICL trials, which amounts to 12K environment steps in total.

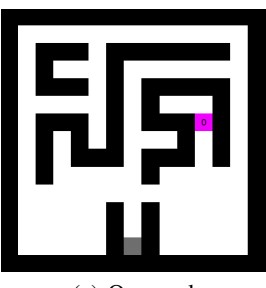 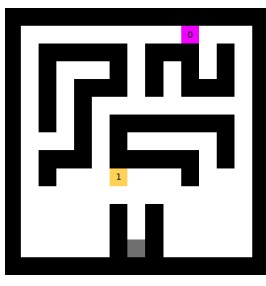 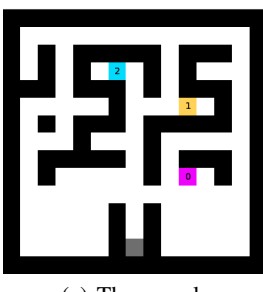

|  (a) One goal | (b) Two goals | (c) Three goals |

**Figure 10:** Maze-Runner environments introduced by Grigsby et al. [2023]. In **Maze-Runner**, the agent is located in a procedurally generated $15 \times 15$ maze and has to navigate to **(a)** one, **(b)** two or **(c)** goal locations in pre-specified order. The agent receives a reward of +1 for reaching a goal. Episodes last for a maximum of 400 steps, or terminate early if all goal locations have been visited.

**Training Dataset.** The procedural-generation of the maze and selection of the number of goals is controlled by setting the environment seed. We use PPO to generate 100K environment interactions for 100 procedurally-generated mazes, and record the entire replay buffer, which amounts to 10M transitions in total. We found it necessary, to equip the task-specific PPO agents with an LSTM [Hochreiter & Schmidhuber, 1997] policy. Without the LSTM, agents hardly make progress for some mazes, especially if the maze contains two or three goal locations. For this reason, we first generate data for more than 100 mazes and select the first 100 seeds, where the average reward at the end of training is $> 0.25$. This results in a set of seeds in $[0, 120]$ Otherwise, we use standard hyperparameter settings as provided in `stable-baselines3`.

**Source Algorithm performance.** We show the average learning curves over all 100 task-specific PPO agents in Figure 11. On average, the agents receive a reward of $\approx 1$ over all mazes. This average include environments with one, two or three goals. We provide further dataset statistics for MazeRunner with the corresponding dataset release.

### B.3 META-WORLD

The Meta-World benchmark [Yu et al., 2020a] consists of 50 challenging robotics tasks, such as opening/closing a window, using a hammer, or pressing buttons. All tasks in Meta-World use a Sawyer robotic arm simulated using the MuJoCo physics engine [Todorov et al., 2012]. The observations and actions are 39-dimensional and 6-dimensional continuous vectors, respectively. As all tasks share the robotic arm, the state, and action spaces remain constant across tasks. All actions are in range $[-1, 1]$. The reward functions are dense and based on distances to the goal locations (exact reward-definitions are provided in Yu et al. [2020a]). Similar to Wolczyk et al. [2021] and Schmied et al. [2024], we limit the episode lengths to 200 interactions. We follow Yu et al. [2020a] and split the 50 Meta-World tasks into 45 training tasks (ML45) and 5 evaluation tasks (ML5). During evaluation, we use deterministic environment resets after episodes, i.e., objects and goal positions are reset to

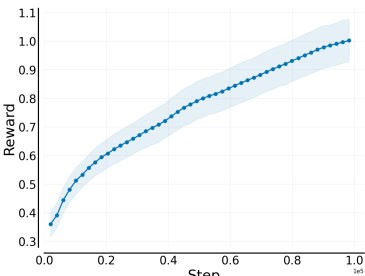

**Figure 11:** Learning curves for data-collection runs on all 100 mazes on Maze-Runner 15×15 environments with PPO-LSTM as source algorithm. We train for 100K environment steps on each maze and report the mean reward achieved (+ 95% CI).

their original state. Furthermore, we mask-out the goal positions in the state vector, which forces agents to adapt during environment interaction. Agents are given 30 ICL trials during evaluation. The 5 evaluation tasks are:

        bin-picking, box-close, door-lock, door-unlock, hand-insert

**Training Dataset**. For our Meta-World experiments, we leverage the datasets released by Schmied et al. [2024]. The datasets contain 2M transitions per task, which amounts to 90M transitions across all ML45 training tasks. The data was generated with randomized object and goal positions after every episode.

### B.4 DMCONTROL

DMControl contains 30 different robotic tasks with different robot morphologies [Tassa et al., 2018]. Similar to prior work [Hafner et al., 2019; Schmied et al., 2024], we select 16 of these 30 tasks and split them into 11 training (DMC11) and 5 evaluation tasks (DMC5). The DMC11 training tasks are:

finger-turn_easy,   fish-upright,   hopper-stand,   point_mass-easy,
walker-stand,   walker-run,   ball_in_cup-catch,   cartpole-swingup,
cheetah-run, finger-spin, reacher-easy

The DMC5 evaluation tasks are:

cartpole-balance,   finger-turn_hard,   pendulum-swingup,   reacher-hard,
walker-walk

States and actions in DMControl are continuous vectors. As DMControl contains different robot morphologies, the state, and action spaces vary considerably across tasks ($3 \leq |\mathcal{S}| \leq 24, 1 \leq |\mathcal{A}| \leq 6$). All actions in DMControl are bounded by $[-1, 1]$. Episodes last for 1000 environment steps and per time-step a maximum reward of +1 can be achieved, which results in a maximum reward of 1000 per episode. Agents are given 30 ICL trials per task during evaluation, which results in 30K steps for a single evaluation run.

**Training Dataset**. As for Meta-World, we leverage the datasets released by Schmied et al. [2024]. The datasets contain 1M transitions per task, which amounts to 11M transitions used for training across all DMC11 tasks. We refer to Schmied et al. [2024] for further dataset statistics on DMControl and Meta-World.

### B.5 PROCGEN

The Procgen benchmark consists of 16 procedurally-generated video games and was designed to test the generalization abilities of RL agents [Cobbe et al., 2020]. Unlike other environments considered in this work, Procgen environments emits $3 \times 64 \times 64$ images as observations. All 16 environments share a common action space of 15 discrete actions. The procedural generation in Procgen is controlled by setting an environment seed. The environments seed randomizes the background and colour of the environment, but retains the same game dynamics. This results in visually diverse observations for

the same underlying task, as illustrated in Figure 12 for three seeds on the game `starpilot`. The rewards in Procgen can be dense or sparse depending on the environment.

We follow Raparthy et al. [2023] and use 12 tasks for training and 4 tasks for evaluation, which we refer to as PG12 and PG4, respectively. The PG12 tasks are:

`bigfish`, `bossfight`, `caveflyer`, `chaser`, `coinrun`, `dodgeball`, `fruitbot`, `heist`, `leaper`, `maze`, `miner`, `starpilot`

The PG4 tasks are: `climber`, `ninja`, `plunder`, `jumper`

We exploit the procedural generation of Procgen and evaluate all models in three settings: (1) training tasks - seen seed (PG12-Seen), (2) training tasks - unseen seed (PG12-Unseen), and (3) evaluation tasks - unseen seed (PG4). In particular, the agents observe data from 200 different training seeds. To enable ICL to the same environment, we always keep the same seed during evaluation (seed=1 for PG12-seen, seed=200 for PG12-Unseen and PG4). During evaluation, we limit the episode lengths to 400 steps.

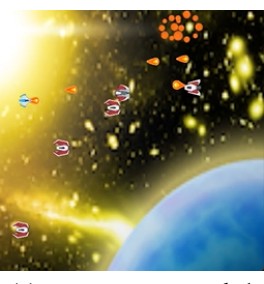 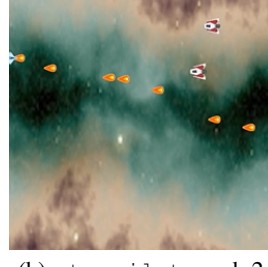 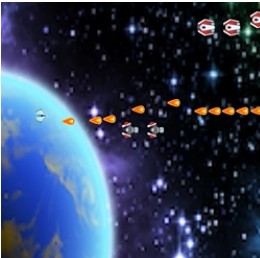

(a) `starpilot`, seed=1      (b) `starpilot`, seed=2      (c) `starpilot`, seed=3

**Figure 12:** Illustration of procedural generation in Procgen `starpilot`. For different seeds, the same environment looks visually considerably different. We train on multi-task dataset of 12 Procgen tasks, with each dataset containing trajectories from 200 environment seeds. To test for ICL, we evaluate on single hold-out seeds.

**Training Dataset.** We generate datasets by training task-specific PPO agents for 25M timesteps on 200 environment seeds per task in `easy` difficulty, as proposed in by Cobbe et al. [2020]. We train PPO using the same hyperparameter settings as Cobbe et al. [2020], using a learning rate of $5e^{-4}$, batch size 2048, number of update epochs of 3, entropy coefficient of 0.01, GAE $\lambda = 0.95$, and with reward normalization. We use 256 timesteps per rollout over 64 parallel environments, which results in 16384 environment steps per rollout in total. Furthermore, we found it useful to decrease the discount factor to 0.99.

As in previous experiments, we record the entire replay buffer and consequently, the datasets contain mixed-quality behaviour. We subsample the 25M transitions per task, by storing only the observations of the first 5 parallel environments, which results in approximately 2M transitions per task. To ensure disk-space efficiency, all trajectories are stored in separate `hdf5` files in the lowest compression level files, with all image-observations encoded in `unit8`. Consequently, the datasets for all 16 tasks (32M transitions) take up only 70GB of disk space, and their `hdf5` format enables targeted reading from disk, without loading an entire trajectory into RAM. We release two versions of our datasets: a smaller one containing 2M transitions per task as used in our experiments, and a larger one containing 20M transitions per task.

**Source Algorithm performance.** We show the individual learning curves for all tasks in Figure 13, and the aggregate statistics over all 16 datasets in Table 1.

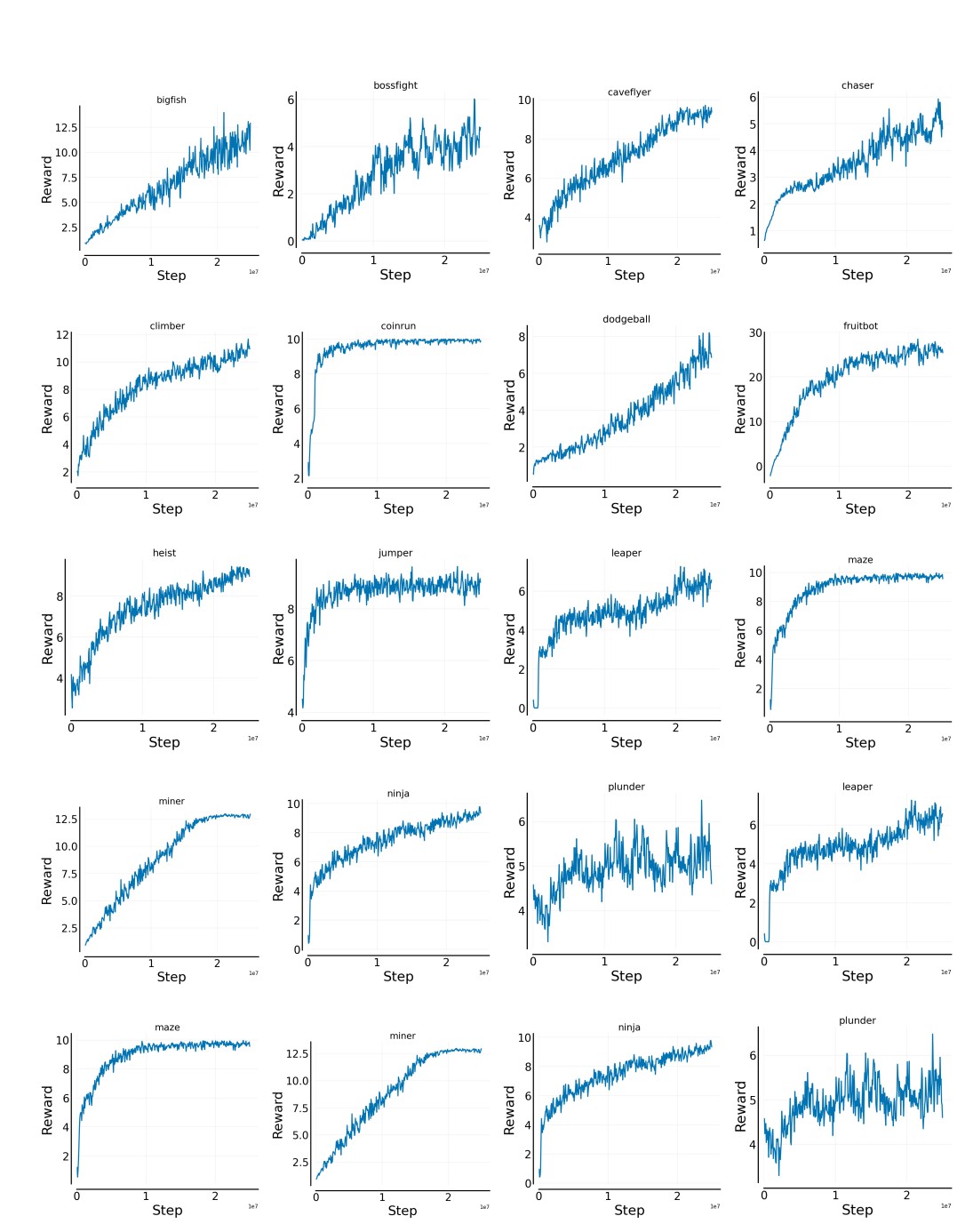

**Figure 13:** Learning curves for data-collection runs on all 16 Procgen environments with PPO as source algorithm. We train for 25M environment steps on each task in `easy` mode.

**Table 1:** Dataset Statistics for all 16 Procgen tasks.

| Task | # of Trajectories | Mean Length | Mean Return |
|------|-------------------|-------------|-------------|
| bigfish | 8834 | $221 \pm 184$ | $5.9 \pm 9.1$ |
| bossfight | 12103 | $161 \pm 200$ | $2.2 \pm 4.3$ |
| caveflyer | 16466 | $119 \pm 202$ | $7.6 \pm 4.4$ |
| chaser | 9182 | $213 \pm 72$ | $3.4 \pm 3.2$ |
| climber | 11392 | $171 \pm 248$ | $9.2 \pm 5.2$ |
| coinrun | 38236 | $51 \pm 49$ | $9.7 \pm 1.8$ |
| dodgeball | 13089 | $149 \pm 214$ | $3.2 \pm 4.2$ |
| fruitbot | 6966 | $280 \pm 152$ | $17.0 \pm 14.3$ |
| heist | 8090 | $241 \pm 395$ | $8.0 \pm 4.0$ |
| jumper | 45621 | $43 \pm 143$ | $8.7 \pm 3.3$ |
| leaper | 28383 | $69 \pm 84$ | $4.9 \pm 5.0$ |
| maze | 48867 | $40 \pm 112$ | $9.5 \pm 2.3$ |
| miner | 26897 | $73 \pm 182$ | $11.7 \pm 3.5$ |
| ninja | 24268 | $80 \pm 136$ | $7.8 \pm 4.2$ |
| plunder | 6179 | $316 \pm 106$ | $4.9 \pm 3.2$ |
| starpilot | 9490 | $206 \pm 137$ | $17.3 \pm 16.4$ |
| Average | 19628 | 152 | 8.2 |

## C    EXPERIMENTAL & IMPLEMENTATION DETAILS

### C.1    GENERAL

**Training & Evaluation.** We compare RA-DT against DT, AD, and DPT on all environments. On grid-world environments, we train all methods for 100K steps and evaluate after every 25K steps. For Meta-World, DMControl and Procgen, we train for 200K steps and evaluate after every 50K steps. During evaluation, the agent is given 40 interaction episodes for ICL on Dark-Room and Dark Key-Door, and 30 episodes on MazeRunner, Meta-World, DMControl, and Procgen. We use the ICL curves as the primary evaluation mechanism, and report the scores at the last evaluation step (100K or 200K). Following, Agarwal et al. [2021] we report the mean and 95% confidence intervals across tasks and over 3 seeds in all experiments.

Across experiments, we keep most parameters fixed, unless mentioned otherwise. We train with a batch size of 128 on all environments, except for 40 grids, where we use a batch size of 32. We use a constant learning rate of $1e^{-4}$ and 4000 linear warm-up steps followed by a cosine decay to $1e^{-6}$ and train using the AdamW optimizer [Loshchilov & Hutter, 2018]. Furthermore, we employ gradient clipping of 0.25, weight decay of 0.01, and a dropout rate of 0.2 for all methods.

**Context Length.** On grid-worlds, we use a context length $C$ equivalent to two 2 episodes for AD, DPT and DT. For example, on $40 \times 20$ grids, this results in a sequence length of 6400 ($= 1600 * 4$ for state/action/reward/RTG) for the DT and a sequence length of 4800 for AD. On Meta-World, DMControl and Procgen, we reduce the sequence context length to 50 steps for DT. For RA-DT, we use a shorter context length of $C = 50$ transitions across environments, except for $20 \times 20$ and $40 \times 20$ grids, where we increase the context length to 100. We want to highlight, that the context length for RA-DT applies to both the input context and the retrieved context. The retrieved context contains the past, and future context, as described in Section 3.2.1. Consequently, the effective context length of RA-DT is $C + 2 * C$ and is independent of the episode length.

**Network Architecture.** For all environments, except for Procgen, we use a GPT2-like network architecture [Radford et al., 2019] with 4 Transformer layers, 8 head and hidden dimension of 512, which results in 16M parameters. On Procgen, we use a larger model with 6 Transformer blocks, 12 heads and hidden dimension of 768. States, actions, rewards and RTGs are embedded using separate embedding layers per modality, as proposed by Chen et al. [2021]. For all modalities and environments, we use standard linear layers to embed the inputs. Procgen is again an exception, where we use the convolutional architecture proposed by Espeholt et al. [2018] and adopted in prior works [Cobbe et al., 2020; Schmidt & Schmied, 2021; Schwarzer et al., 2023]. Processing image-sequences is computationally demanding. Therefore, we first pre-train the vision-encoder using a separate DT and embed all images in the dataset using the learned vision encoder. Therefore, the data-loading is not bottlenecked by loading entire images into memory, but only their compact representations.

Furthermore, we use global positional embeddings. We also experimented with the Transformer++ recipe (RoPE, SwiGLU, RMSNorm), but only observed minimal performance gains for our problem setting. To speed-up training, we use mixed-precision Micikevicius et al. [2017], model compilation as supported in PyTorch [Paszke et al., 2019], and FlashAttention [Dao, 2023].

**Implementation.** Our implementation of the DT is based on the `transformers` library [Wolf et al., 2020] and `stable-baselines3` [Raffin et al., 2021]. We integrated AD, DPT, and RA-DT on top of this implementation.

**Hardware & Training Times.** We run all our experiments on a server equipped with 4 A100 GPUs. For most of our experiments, we only use a single A100. Depending on the environment and method used, training times range from one hour (Dark-Room, DT) to 20 hours (DMControl, AD) for a single training run.

### C.2    DECISION TRANSFORMER

For Dark-Room and Dark Key-Door, we sample the target return for RTG conditioning before every episode $\mathcal{N}(90, 5)$, $\mathcal{N}(370, 10)$, and $\mathcal{N}(500, 10)$ for grid sizes $10 \times 10$, $20 \times 20$, and $40 \times 20$, respectively. On grid-worlds, we found that sampling the target return performs better than using a fixed target return per grid size. We assume this is, because specifying a particular target return

biases the DT towards particular goal locations. For MazeRunner, we use a constant target return of 3. For Meta-World, DMControl, and Procgen, we set the target return the maximum return achieved for a particular task in the training datasets. However, we also found that constant target returns per domain work decently.

## C.3 Algorithm Distillation

AD obtains a context trajectory and learns to predict actions of an input trajectory taken $K$ episodes later. Therefore, we tune $K$ per domain. On grid-worlds, we found $K = 100$ to perform the best, similar to Lee et al. [2023]. For MazeRunner and Meta-World, we set $K = 1000$, and for DMControl and Procgen, we set $K = 250$.

## C.4 Retrieval-Augmented Decision Transformer

**Embedding Model.** For the embedding model $g(\cdot)$, we either use a DT pre-trained on the same environment with the same hyperparameters as listed in Section C, or a pre-trained and frozen LM. For the pre-trained LM, we use `bert-base-uncased` from the `transformers` library by default. BERT is an encoder-only LM with 110M parameters, vocabulary size $v = 30522$, and embedding dimension of $d_{LM} = 768$ [Devlin et al., 2019]. We apply FrozenHopfield with $\beta = 10$ to state, action, reward and RTG tokens (see Equation 2). To achieve this, we one-hot encode all discrete input tokens, such as actions in Dark-Room/MazeRunner/Procgen or states in Dark-Room, and rewards/RTGs in the sequence before applying the FH. For other tokens, such as continuous states/actions as in Meta-World/DMControl, we directly apply the FH. We evaluate other alternatives for the LM in Appendix E.

**Constructing queries/keys/values.** Regardless of whether $g$ is domain-specific or domain-agnostic, we obtain $C$ embedded tokens after applying $g$ to the input trajectory $\tau_{in}$. Subsequently, we apply mean aggregation over the context length $C$ to obtain the $d_r$-dimensional query representation. We experimented with aggregating over all tokens or only tokens of a particular modality (state/action/reward/RTG), and found aggregation over states-only to be most effective (see Appendix E.4). As described in Section 3.2.1, we construct the key-value pairs in our retrieval index by embedding all sub-trajectories in the dataset $\mathcal{D}$ using our embedding model $g$, $\mathcal{K} \times \mathcal{V} = \{(g(\tau_{i,t-C:t}), \tau_{i,t-C:t+C}) \mid 1 \leq i \leq |\mathcal{D}|\}$. To avoid redundancy, in practice we construct $H/C$ key-value pairs for a given trajectory $\tau$ with episode length $H$ and sub-sequence length $C$, instead of constructing the key and values for every step $t \in [1, H]$. Note that the values, we store $\tau_{i,t-C:t+C}$, contain both the sub-trajectory itself ($\tau_{i,t-C:t}$) and its continuation ($\tau_{i,t:t+C}$). Similar to Borgeaud et al. [2022], we found this choice important for high performance in RA-DT, because it allows the model to observe how the trajectory may evolve if it predicts a certain action (given that the retrieved context is similar enough).

**Vector Index.** We use Faiss [Johnson et al., 2019; Douze et al., 2024] to instantiate our vector index $\mathcal{I}$. This allows us to search our vector index in $\mathcal{O}(\log M)$ time using Hierarchical Navigable Small World (HNSW) graphs. However, in practice we found it faster to use a Flat index on the GPU as provided by Faiss instead of using HNSW, because our retrieval datasets are small enough. We use retrieval both during training and during inference. It is, however, possible to pre-compute the retrieved trajectories for $\mathcal{D}$ prior to the training phase to limit the computational demand of retrieval, as suggested by Borgeaud et al. [2022]. During evaluation, we can retrieve after every environment step or only after every $t$ environment steps. Here, $t$ represents a trade-off between inference time and final performance. We use $t = 1$ for Dark-Room and Dark Key-Door, and $t = 25$ for all other environments (see Appendix E.6 for an ablation on this design choice). For all environments, except for Meta-World and DMControl, we provide a single retrieved sub-trajectory in the agent's context. For Meta-World and DMControl, we found that providing more than one retrieved sub-trajectory benefits the agent's performance. Therefore, for these two environments, we retrieve the top-4 sub-trajectories, order them by return achieved in that trajectory, and provide their concatenation as retrieved context for RA-DT.

**Reweighting.** To implement the reweighting mechanism, as described in Section 3.2.3, we first retrieve the top $l \gg k$ experiences and the select the top-$k$ experiences according to their reweighted scores. We set $l = 50$ in all our experiments.

**Embedding Retrieved Context.** After the most similar trajectories have been retrieved, we embed the state/action/reward/RTG tokens with a separate embedding layers (as is done for the regular input sequence) before incorporating them via the CA layers. We also experimented with sharing/detaching the regular embedding layers, but found it most effective to maintain separate ones. Furthermore, we experimented with an additional Transformer-based encoder for the retrieved sequences, as proposed by Borgeaud et al. [2022], but did not observe substantial performance gains despite increased computational cost.

**Retrieval Dataset.** For all our experiments, we use the same dataset for retrieval $\mathcal{D}'$ as is used for training $\mathcal{D}$, that is $\mathcal{D}' = \mathcal{D}$. Therefore, we prevent retrieving sub-sequences from the same trajectory as the query.

**Retrieval Regularization.** We found it advantageous to regularize the k-NN retrieval in RA-DT throughout the training phase. In RL datasets, there is often a substantial overlap between trajectories, leading to many similar sub-trajectories. This poses a significant challenge, as retrieving only similar sub-trajectories encourages the agent to adopt copying behaviour, which renders the DT unable to produce high-reward actions during inference.

One simple strategy to mitigate this issue is **deduplication**, i.e., to discard duplicate experiences before the training phase of RA-DT. To achieve this, we first construct our index as described in Section 3.2. For every key $\mathbf{k} \in \mathbf{K}$, we retrieve the top-$k$ neighbours (excluding experiences from the same episode as $\mathbf{k}$). If the similarity score is above a cosine similarity of $0.98$, we discard the experience. This substantially reduces the number of experiences in the index and speeds-up retrieval.

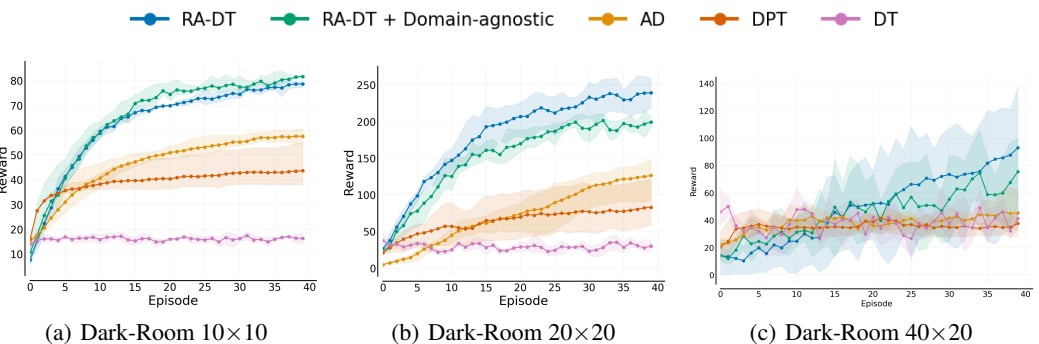

(a) Dark-Room 10×10     (b) Dark-Room 20×20     (c) Dark-Room 40×20

**Figure 14:** In-context learning performance on **(a) Dark-Room 10×10**, **(b) Dark-Room 20×20**, **(c) Dark-Room 40×20** at end of training (100K steps). We evaluate each agent for 40 episodes on each of the 80 **training tasks** and report mean reward (+ 95% CI) over 3 seeds.

Two other strategies for regularizing retrieval during the training phase, are **similarity cut-off** and **query dropout** [Yasunaga et al., 2023]. Similarity cut-off first retrieves the top $m > l$ experiences, discards the experiences with a similarity score above a threshold (e.g., $0.98$), and retains only the remaining experiences $l$. If used in combination with reweighting, we set $m = 2 * l$. Query dropout randomly drops-out tokens (e.g., 20%) of the embedded sub-trajectory $\tau_{\text{in}}$, which leads to more diverse retrieved experiences. We found both strategies effective for RA-DT. We use query dropout of 0.2, similarity cut-off of 0.98, and deduplication by default. Furthermore, for Meta-World and DMControl, we found **query-blending** useful. Query-blending interpolates between then actual query and a randomly selected key from the retrieval index, $\boldsymbol{q}' = \boldsymbol{q} * \alpha_{\text{blend}} + (1 - \alpha)\boldsymbol{q}_{\text{rand}}$. For Meta-World and DMControl we additionally set $\alpha_{\text{blend}} = 0.5$.

On Dark-Room and Dark Key-Door environments, we found it useful to replace retrieved experiences with experiences randomly sampled from the same task, if the query sub-sequence is from the beginning of the episode (i.e., smaller than timestep 10). This is because on these two environments, retrieving appropriate experience can be difficult if the given query sub-sequence is too short.

Finally, we use the same RTG-conditioning strategy as the vanilla DT, as described in Appendix C.2.

---

**Algorithm 2** RA-DT at training time

---

**Input:** DT $\pi_\theta$, embed model $g$, dataset $\mathcal{D}$, gradient steps $N$, context len $C$, batch size $B$, eval frequency $E$, loss function $\mathcal{L}$ (cross-entropy or MSE), `evaluate`, batch-wise procedures `retrieve`, `reweight`, and `update`

1: $\mathcal{I} \leftarrow \emptyset$              ▷ Initialize retrieval index $\mathcal{I}$
2: **for** $\tau \in \mathcal{D}$ **do**
3:   $\mathcal{I} \leftarrow \mathcal{I} \cup \{(g(\tau_{t-C:t}), \tau_{t-C:t+C}) \mid t \in \text{range}(0, |\tau|, C)\}$   ▷ Add k-v pairs of sub-trjs to $\mathcal{I}$
4: **end for**
5: **for** $i = 1 \ldots N$ **do**
6:   $\mathbf{b} \sim \mathcal{D}$ where $\mathbf{b} = \{\tau_j \mid 1 \le j \le B\}$     ▷ Sample batch of sub-trjs each of length $C$
7:   $\mathbf{q} = g(\mathbf{b})$             ▷ Construct queries for all sub-trjs
8:   $\mathcal{R} \leftarrow \texttt{retrieve}(\mathbf{q}, \mathcal{I})$        ▷ Retrieve top-$l$ sub-trjs, Eq. 3
9:   $\mathcal{S} \leftarrow \texttt{reweight}(\mathcal{R})$        ▷ Re-weight top-$k$ sub-trjs, Eq. 4, 5
10:   $\mathbf{a} = \pi_\theta(\cdot \mid \mathbf{b}, \{\tau_{\text{ret}} \in \mathcal{S}\})$       ▷ Predict actions for batch
11:   $\pi_\theta \leftarrow \texttt{update}(\pi_\theta, \mathcal{L}, \mathbf{a}, \mathbf{b})$    ▷ Perform gradient step, see Appendix C.1 for $\mathcal{L}$
12:   **if** $i \% E == 0$ **then**
13:    $\texttt{evaluate}(\pi_\theta, g)$        ▷ Evaluation with ICL, see Algorithm 1
14:   **end if**
15: **end for**

---

# D ADDITIONAL RESULTS

## D.1 DARK-ROOM

Analogous to the ICL curves on the 20 evaluation tasks in Figure 3, we present ICL curves on the 80 train tasks in Figure 14. In general, we observe a similar learning behaviour on the train tasks as on the evaluation tasks, with slightly higher scores on average. Interestingly, the domain-agnostic variant of RA-DT slightly outperforms its domain-specific counterpart on the training tasks.

In addition, we also show the learning curves on Dark-Room $10 \times 10$ over the entire training phase in Figure 15. We evaluate after every 25K updates and observe a steady improvement in the average performances with every evaluation.

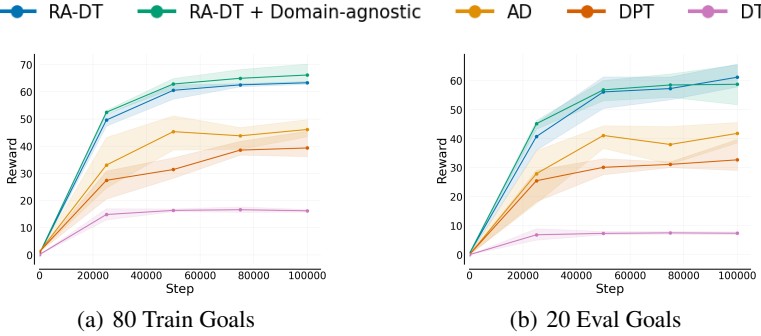

(a) 80 Train Goals          (b) 20 Eval Goals

**Figure 15:** Average performances on **Dark-Room 10×10** over the course of training for **(a)** train and **(b)** test tasks. We train each agent for 100K steps and evaluate every 25K steps. Curves are averaged across the 80 train and 20 evaluation tasks, respectively. We report mean reward (+ 95% CI) over 3 seeds.

### D.1.1 ATTENTION MAP ANALYSIS

We conduct a qualitative analysis on Dark-Room $10 \times 10$ to better understand how RA-DT leverages the retrieved context sub-sequences. First, we analyse the attention maps for different Dark-Room $10 \times 10$ goal locations.

**Table 2:** Hyperparameters for RA-DT.

| Environment | Parameter | Value |
|---|---|---|
| Default | Gradient steps | 100K |
| | Optimizer | AdamW |
| | Batch size | 128 |
| | Lr schedule | Linear warm-up + Cosine |
| | Warm-up steps | 4000 |
| | Learning rate | 1e-4 $\rightarrow$ 1e-6 |
| | Weight decay | 0.01 |
| | Gradient clipping | 0.25 |
| | Dropout | 0.2 |
| | Context Length | 50 timesteps |
| | Top-$k$ before re-weighting | 50 |
| | Top-$k$ after re-weighting | 1 |
| | Eval steps between retrievals | 1 |
| | Query sequence aggregation | mean |
| | Query sequence tokens | state |
| | Query dropout | 0.2 |
| | Re-weight $\alpha$ | 1 |
| | Train re-weighting | task |
| | Eval re-weighting | return |
| | Similarity cut-off | 0.98 |
| | Deduplicate | True |
| | Min len for retrieval (only for Dark) | 10 |
| | Domain-agnostic LM | `bert-base-uncased` |
| | Domain-agnostic LM hidden dim | 768 |
| | FrozenHopfield $\beta$ | 10 |
| Dark Room/Key-Door $20 \times 20$ | Context length | 100 |
| Dark Room/Key-Door $40 \times 20$ | Context length | 100 |
| | Batch size | 32 |
| MazeRunner | Eval steps between retrievals | 25 |
| Meta-World/DMControl | Gradient steps | 200K |
| | Eval steps between retrievals | 25 |
| | Top-$k$ after re-weighting | 4 |
| | Query blending | 0.5 |
| Procgen | Gradient steps | 200K |
| | Eval steps between retrievals | 25 |

**What happens if an optimal trajectory is retrieved in context?** In Figure 16, we showcase this example. The goal location is located at grid cell (4,6). The attention maps exhibit high attention scores for the state and the RTG at the end of the retrieved trajectory. We also observe high attention scores for the state similar to the current state and the action selected in that state. The agent initially imitates the actions in the context trajectory, but deviates further into the episode. Once the agent reaches the goal state, the attention scores for states and RTGs at the end of the trajectory reduce considerably, because the agent need not pay attention to the retrieved context any more.

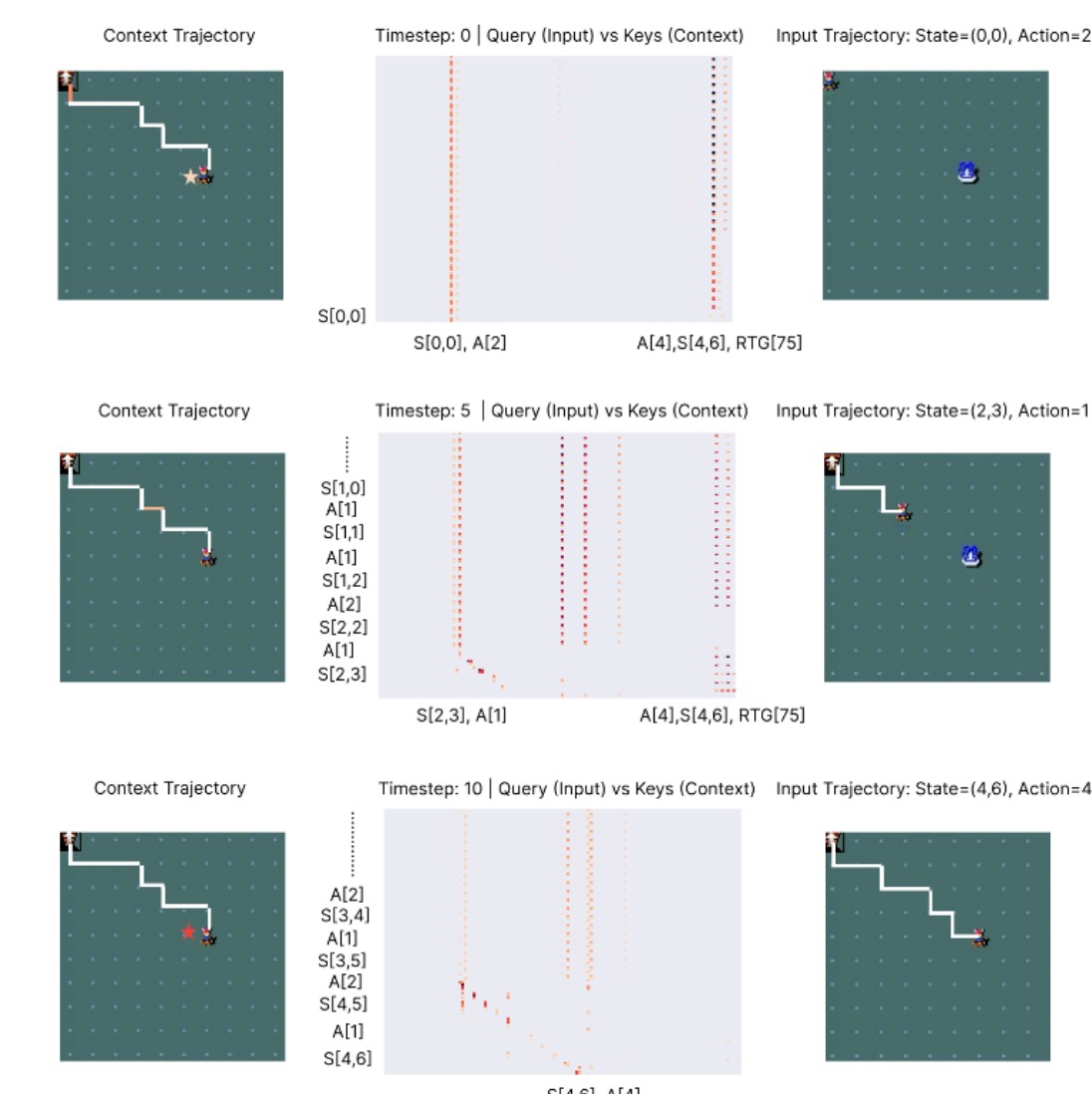

**Figure 16:** Attention map analysis for an **optimal context-trajectory** on Dark-Room $10 \times 10$. We plot the retrieved context trajectory (left), the corresponding attention map, and actual agent state (right), across timesteps (1, 5, 10). Queries (input trajectory) are on the y-axis and keys (context trajectory) on the x-axis. We highlight the sub-sequence in the context trajectory with the highest attention score (left). To improve readability, we mask-out attention scores below a certain threshold, and only provide labels for token that exhibit the highest attention scores. The agent imitates the context trajectory and successfully finds the goal.

**What happens if a suboptimal trajectory is retrieved in Context?** Similarly, we show the corresponding example in Figure 17. The goal location is again in grid cell (4,6). The retrieved context trajectory reaches the final state (9,5). Similar to Figure 16, the attention maps exhibit high

attention scores for the last state and RTG for that state, as well as for a state at a similar timestep. Previously, RA-DT imitated the action, but in this situation the agent picks a different route, as the context trajectory does not lead to a successful outcome.

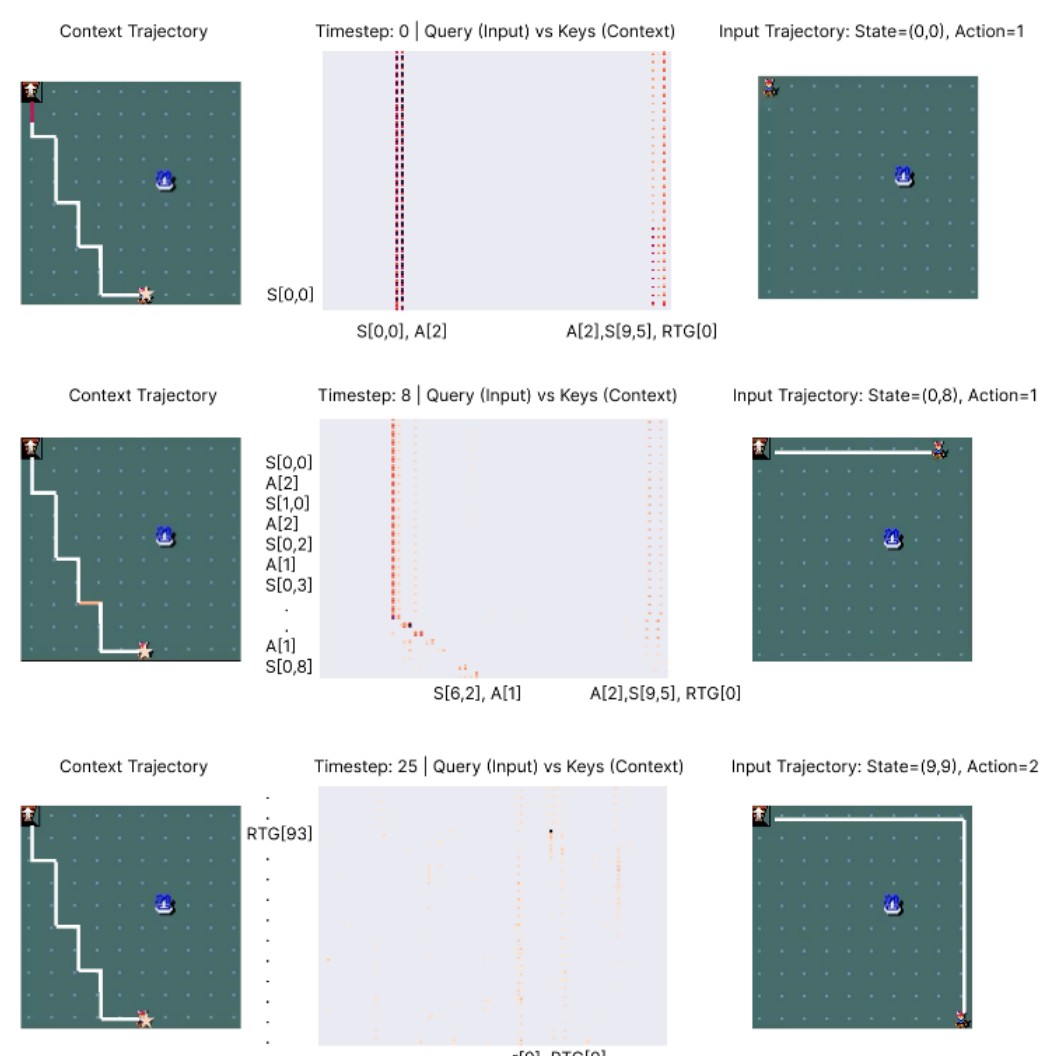

**Figure 17:** Attention map analysis for a **suboptimal context-trajectory** on Dark-Room $10 \times 10$. The agent selects a different route than present in the suboptimal context trajectory and explores the environment.

This analysis suggests, that RA-DT can develop capabilities to either imitate a given positive experience or to behave differently than a given negative experience.

### D.1.2 EXPLORATION ANALYSIS

**State Visitations.** In Section D.1.1, we found that RA-DT learned to either copy or avoid behaviours given positive or negative context trajectories. Therefore, we further analyse the exploration behaviour of RA-DT by visualizing the state-visitation frequencies on Dark-Room $10 \times 10$ across the 40 ICL trials for three different goal locations: $(5, 8)$, $(5, 1)$, and $(4, 6)$ (see Figure 18). The agent visits nearly all states at least once at test time, as visualized in Figure 18 (a) and (b). Once the agent finds the goal location, it starts to imitate and stops exploring, as illustrated in Figure 18 (c).

**Delusions in RA-DT.** Furthermore, we find that in some unsuccessful trials, the agent repeatedly performs the same suboptimal action sequences. Ortega et al. [2021] refer to such behaviour as

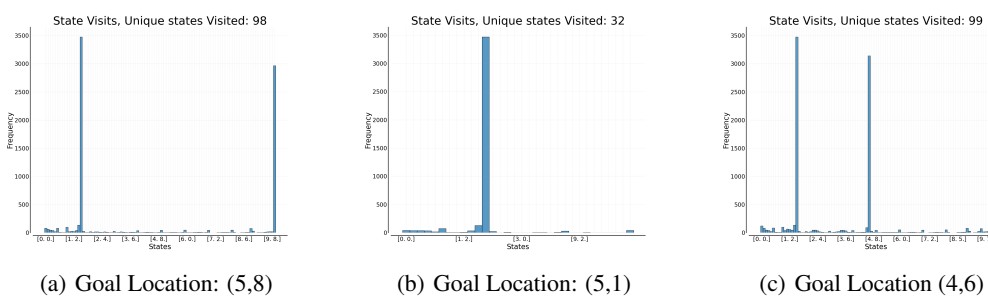

(a) Goal Location: (5,8)    (b) Goal Location: (5,1)    (c) Goal Location (4,6)

**Figure 18:** We count the state visitations on Dark-Room $10 \times 10$ over all ICL trials for three different goal locations: $(5, 8)$, $(5, 1)$, and $(4, 6)$. The total number of states is 100. The agent attempts to visit all states at least once. Once the agent finds the goal, it starts exploiting (e.g., goal location $(5, 1)$).

delusions. In Figure 19, we illustrate two examples in which the agent suffers from delusions and does not recover until the end of the episode.

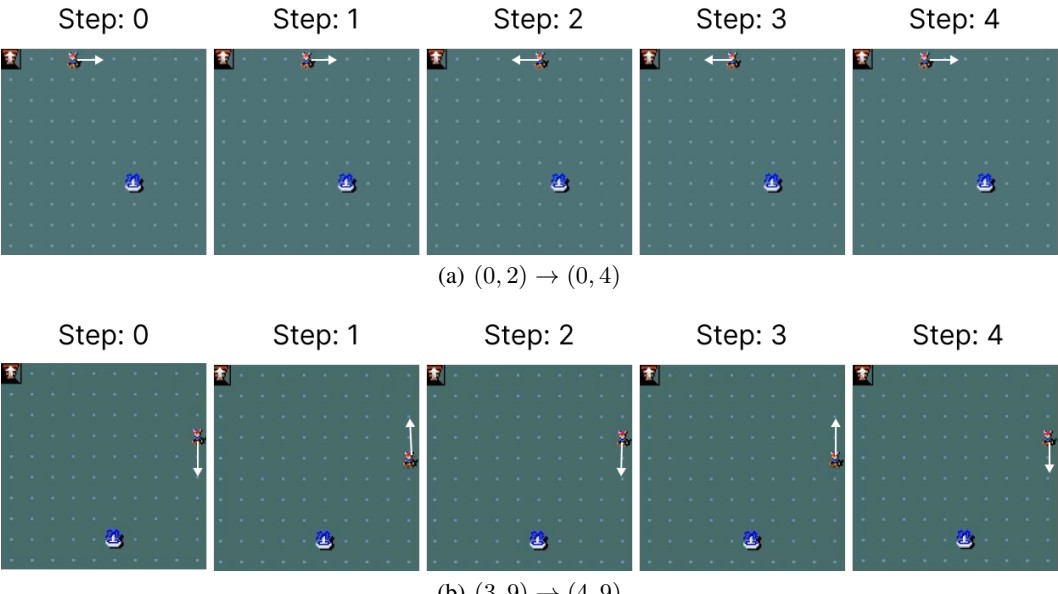

**Figure 19:** Illustrations of delusions in RA-DT on Dark-Room $10 \times 10$. In **(a)**, the agent navigates from state (0, 2) to (0, 4) and returns to (0, 2). In **(b)**, the agent The agent goes from state (3, 9) and (4, 9) and back. In both examples, the agent repeats the unsuccessful action sequence.

## D.2 MAZE-RUNNER

In Figures 20 and 21, we report the average performances at the end of the training (100K) for both the 100 train and 20 evaluation mazes, as well as the corresponding ICL curves, respectively.

While RA-DT outperforms competitors, we observe a considerable performance gap between train mazes and test mazes (0.65 vs. 0.4 reward, see Figure 20). This indicates that RA-DT struggles to solve difficult, unseen mazes. We believe that this gap is an artifact of the small pre-training distribution of 100 mazes, and be closed by increasing the number of pre-training mazes. Furthermore, increasing the number of ICL trials may also enhance the performance.

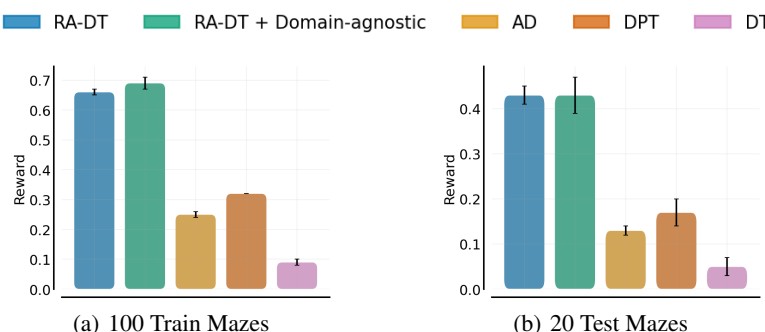

**Figure 20:** Average performance on **(a)** 100 train and **(b)** 20 test mazes at end of training (100K steps). We evaluate each agent for 30 episodes and report mean reward (+ 95% CI) over 3 seeds.

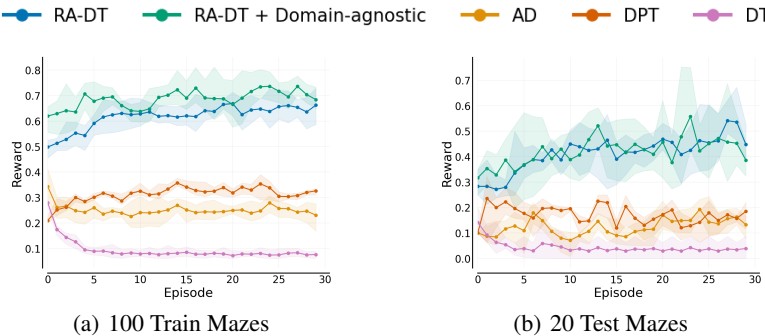

**Figure 21:** ICL on **(a)** 100 train and **(b)** 20 test mazes at end of training (100K steps). We evaluate each agent for 30 episodes and report mean reward (+ 95% CI) over 3 seeds.

### D.3   META-WORLD

In Figures 22 and 23, we show the training curves across the entire training period (200K steps), and the corresponding ICL curves at the end of training for both ML45 and ML5.

Generally, we observe that RA-DT outperforms competitors on the evaluation tasks in terms of average performance. However, on training task, the average performance of RA-DT is lower than of the vanilla DT. AD and DPT lack behind both methods. One potential reason is the RTG conditioning, which biases DT and RA-DT towards higher quality behaviour.

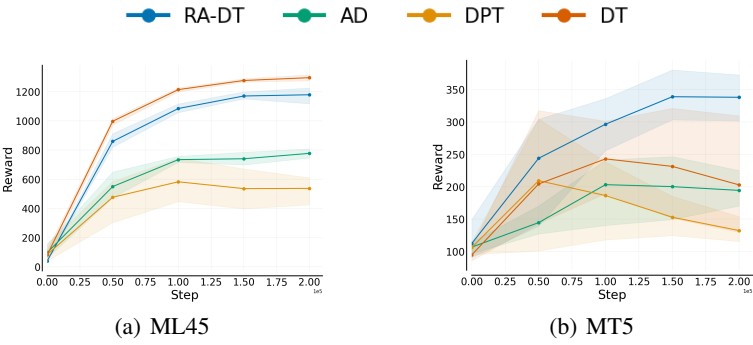

**Figure 22:** Learning curves on **(a)** ML45 and **(b)** MT5 over the full training period (200K). We evaluate each agent for 30 episodes and report mean reward (+ 95% CI) over 3 seeds.

Nevertheless, we do not observe improved ICL performance of RA-DT on evaluation tasks. While all in-context RL methods exhibit in-context improvement on the training tasks (ML45), neither RA-DT nor other methods show signs of improvement on the evaluation tasks (MT5).

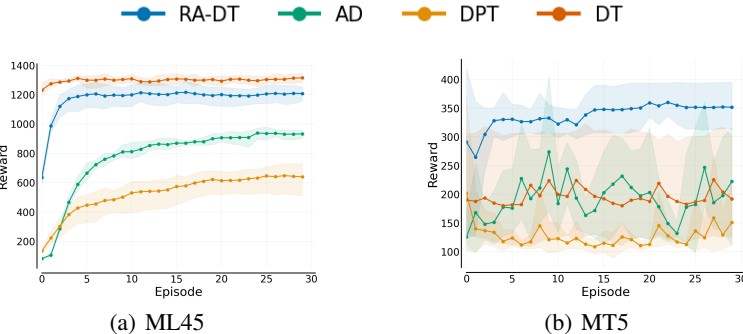

(a) ML45           (b) MT5

**Figure 23:** ICL performance on **(a)** ML45 and **(b)** MT5 at end of training (200K steps). We evaluate each agent for 30 episodes and report mean reward (+ 95% CI) over 3 seeds.

In addition, we provide the average rewards and data-normalized scores in for the MT5 evaluation tasks in Table 3.

**Table 3:** Meta-World Evaluation Tasks.

| Environment | DT | AD | DPT | RA-DT |
|---|---|---|---|---|
| **Reward** | | | | |
| bin-picking | 62.28 ± 34.37 | 42.63 ± 17.47 | 27.52 ± 14.07 | 14.47 ± 1.79 |
| box-close | 70.34 ± 6.72 | 85.4 ± 14.96 | 106.79 ± 23.7 | 110.09 ± 46.69 |
| hand-insert | 27.38 ± 3.1 | 51.82 ± 59.93 | 13.06 ± 0.15 | 182.25 ± 99.63 |
| door-lock | 229.76 ± 11.4 | 333.89 ± 161.77 | 239.2 ± 20.19 | 219.44 ± 2.51 |
| door-unlock | 588.66 ± 454.89 | 450.71 ± 8.37 | 249.17 ± 63.38 | 1163.02 ± 36.42 |
| **Average** | 195.68 ± 97.31 | 192.89 ± 23.48 | 127.15 ± 16.97 | 337.85 ± 34.94 |
| **Data-normalized Scores** | | | | |
| bin-picking | 0.24 ± 0.14 | 0.16 ± 0.07 | 0.09 ± 0.06 | 0.04 ± 0.01 |
| box-close | -0.07 ± 0.01 | -0.03 ± 0.03 | 0.01 ± 0.05 | 0.02 ± 0.1 |
| hand-insert | 0.02 ± 0.0 | 0.04 ± 0.05 | 0.01 ± 0.0 | 0.15 ± 0.08 |
| door-lock | 0.0 ± 0.01 | 0.08 ± 0.12 | 0.01 ± 0.01 | -0.0 ± 0.0 |
| door-unlock | 0.27 ± 0.31 | 0.18 ± 0.01 | 0.04 ± 0.04 | 0.66 ± 0.02 |
| **Average** | 0.09 ± 0.08 | 0.08 ± 0.01 | 0.03 ± 0.03 | 0.17 ± 0.04 |

### D.4 DMCONTROL

In Figures 24 and 25, we show the training curves across the entire training period (200K steps), and the corresponding ICL curves at the end of training for both DMC11 and DMC5.

Similar to our results on Meta-World, we observe that RA-DT outperforms competitors on average. However, we do not observe in-context improvement on the evaluation tasks.

In addition, we show the average rewards obtained and corresponding data-normalized scores for all DMC5 evaluation tasks in Table 4.

### D.5 PROCGEN

In Figures 26 and 27, we show the training curves across the entire training period (200K steps), and the corresponding ICL curves at the end of training for PG12-Seen, PG12-Unseen, and PG4. While

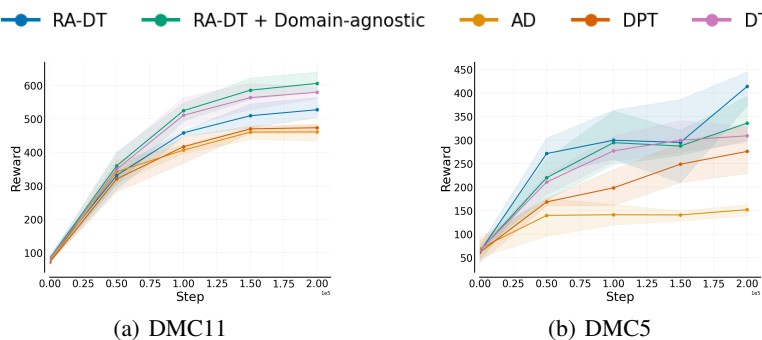

(a) DMC11        (b) DMC5

**Figure 24:** Average performance on **(a)** DMC11 and **(b)** DMC5 at end of training (200K steps). We evaluate each agent for 30 episodes and report mean reward (+ 95% CI) over 3 seeds.

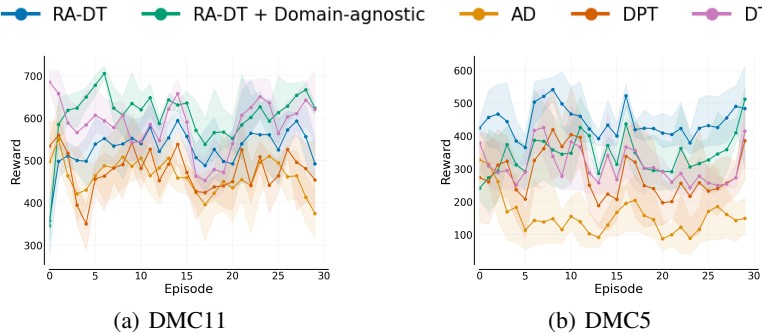

(a) DMC11        (b) DMC5

**Figure 25:** ICL performance on **(a)** DMC11 and **(b)** DMC5 at end of training (200K steps). We evaluate each agent for 30 episodes and report mean reward (+ 95% CI) over 3 seeds.

we observe slightly better average performance of RA-DT compared to competitors, we do not find any in-context improvement.

**RA-DT constructs bursty sequences.**. Building on work by Chan et al. [2022], Raparthy et al. [2023] identified trajectory burstiness as one important property for ICL to emerge on the Procgen benchmark. A given sequence is considered bursty, if it contains at least two trajectories from the same seed (or level). Consequently, the agent obtains relevant information that it can leverage to predict the next action. Therefore, we follow Raparthy et al. [2023] and always provide a trajectory from the same seed in the context of AD and DPT. Indeed, we observed that this improves performance, compared to not taking trajectory burstiness into account. Interestingly, we found that RA-DT retrieves trajectories from the same or similar seeds (seed accuracy of 80%), that is, RA-DT automatically constructs bursty sequences. This intuitively makes sense, as retrieval directly searches for the most relevant experiences (see Section 3.2.3). Therefore, for RA-DT, we do not provide additional information that indicates with which environment seed the trajectory was generated.

# E    ABLATION STUDIES

To better understand the effect of learning with retrieval, we presented a number of ablation studies on critical components in RA-DT (Section 4.6). We conduct all ablations on Dark-Room $10 \times 10$ and otherwise retain the same experiment design choices, as reported in Section 4.1.

## E.1    RETRIEVAL OUTPERFORMS SAMPLING OF EXPERIENCES

RA-DT is conditioned on sub-trajectories via cross-attention. By default, RA-DT leverages retrieval to search for relevant sub-trajectories for a given input sequence. Instead of retrieval, sub-trajectories can be sampled at random from the external memory. Therefore, we conduct an ablation in which we

**Table 4:** DMControl Eval Tasks.

| Environment | AD | DT | DPT | RA-DT |
|---|---|---|---|---|
| **Reward** | | | | |
| cartpole-balance | $211.96 \pm 62.8$ | $946.49 \pm 44.91$ | $703.89 \pm 263.11$ | $910.1 \pm 106.0$ |
| finger-turn_hard | $199.34 \pm 46.0$ | $253.13 \pm 43.0$ | $295.2 \pm 51.88$ | $336.37 \pm 16.51$ |
| pendulum-swingup | $1.18 \pm 2.04$ | $0.0 \pm 0.0$ | $0.0 \pm 0.0$ | $0.0 \pm 0.0$ |
| reacher-hard | $34.22 \pm 17.25$ | $167.7 \pm 42.86$ | $157.29 \pm 94.79$ | $95.4 \pm 15.4$ |
| walker-walk | $326.42 \pm 102.52$ | $189.46 \pm 10.22$ | $257.11 \pm 57.21$ | $877.9 \pm 15.2$ |
| **Average** | $154.63 \pm 12.17$ | $311.36 \pm 12.49$ | $282.7 \pm 54.62$ | $443.95 \pm 25.7$ |
| **Data-normalized Score** | | | | |
| cartpole-balance | $-0.24 \pm 0.11$ | $1.01 \pm 0.08$ | $0.6 \pm 0.45$ | $0.95 \pm 0.18$ |
| finger-turn_hard | $0.25 \pm 0.07$ | $0.33 \pm 0.07$ | $0.4 \pm 0.08$ | $0.47 \pm 0.03$ |
| pendulum-swingup | $0.0 \pm 0.0$ | $0.0 \pm 0.0$ | $0.0 \pm 0.0$ | $0.0 \pm 0.0$ |
| reacher-hard | $0.03 \pm 0.02$ | $0.21 \pm 0.06$ | $0.19 \pm 0.12$ | $0.11 \pm 0.02$ |
| walker-walk | $0.4 \pm 0.14$ | $0.21 \pm 0.01$ | $0.31 \pm 0.08$ | $1.14 \pm 0.02$ |
| **Average** | $0.09 \pm 0.02$ | $0.35 \pm 0.02$ | $0.3 \pm 0.09$ | $0.53 \pm 0.04$ |

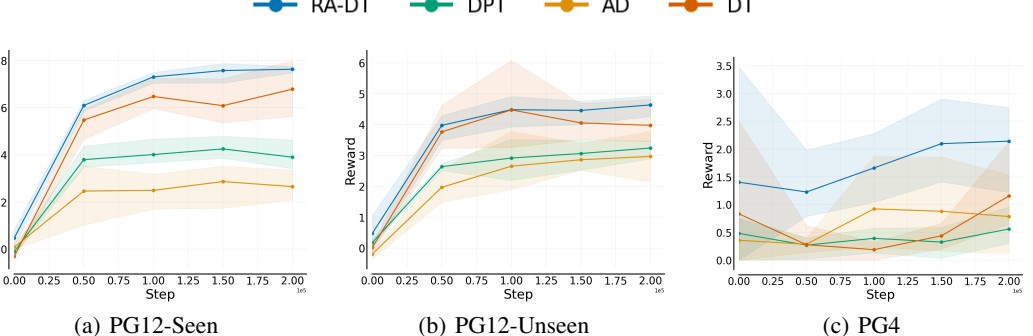

| (a) PG12-Seen | (b) PG12-Unseen | (c) PG4 |

**Figure 26:** Learning curves on Procgen across **(a)** PG12-Seen, **(b)** PG12-Unseen, and **(c)** PG4 seed over the full training period. We train for 200K steps, evaluate every 50K steps for 30 episodes, and report mean reward (+ 95% CI) over 3 seeds.

swap the retrieval mechanism with random sampling of sub-trajectories during training. This is to investigate the effect of relevance of retrieved sub-trajectories on learning performance. We apply random sampling only during training and use our regular retrieval during inference.

In Figure 6a, we show the ICL curves for training RA-DT with retrieved sub-trajectories, sub-trajectories sampled from the same task as the input sequence, and sub-trajectories sampled uniformly across all tasks. We find that training with retrieval outperforms both sampling variants. Uniform sampling results in poor ICL performance. A reason for this, is that context trajectories from a different goal location, are not relevant for predicting actions in the current sequences. As a result, the model ignores the given context during the training phase, and subsequently is unable to leverage it during inference. In contrast, sampling sub-trajectories from the same task as the input sequence results in better ICL performance, as the model learns to make use of the context trajectories. Nevertheless, using retrieval results in even better ICL performance, as sub-trajectories are not only relevant for the current task, but also similar to the current situation.

### E.2 REWEIGHTING MECHANISM

Next, we evaluate how our reweighting mechanism affects the ICL abilities of RA-DT. RA-DT reweights a sub-trajectory by its *relevance* and *utility* score (see Section 3.2). During training, we set $s_u(\tau_{\mathrm{ret}}) = 1$, if the $\tau_{\mathrm{ret}}$ is from the same task as $\tau_{\mathrm{in}}$, and 0 otherwise. Instead of reweighting by task ID, alternatives are to reweight a $\tau_{\mathrm{ret}}$ by its return achieved or by its position in the training dataset.

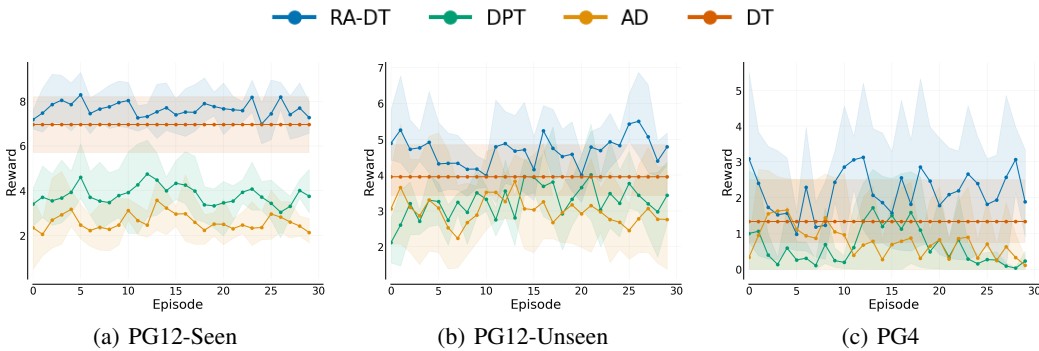

(a) PG12-Seen  (b) PG12-Unseen  (c) PG4

**Figure 27:** ICL performances on Procgen across **(a)** PG12-Seen, **(b)** PG12-Unseen, and **(c)** PG4. We evaluate for 30 episodes, and report mean reward (+ 95% CI) over 3 seeds.

**Table 5:** Procgen Train Tasks, Train Seeds.

| Environment | DT | AD | DPT | RA-DT |
|---|---|---|---|---|
| | | **Rewards** | | |
| bigfish | $4.67 \pm 3.51$ | $2.0 \pm 0.76$ | $2.41 \pm 0.1$ | $5.21 \pm 0.25$ |
| bossfight | $1.0 \pm 0.0$ | $0.46 \pm 0.55$ | $0.9 \pm 0.26$ | $1.31 \pm 0.08$ |
| caveflyer | $3.33 \pm 5.77$ | $0.22 \pm 0.19$ | $3.0 \pm 3.28$ | $9.67 \pm 0.0$ |
| chaser | $1.49 \pm 1.05$ | $1.7 \pm 0.49$ | $1.64 \pm 0.59$ | $2.78 \pm 0.46$ |
| coinrun | $6.67 \pm 5.77$ | $5.89 \pm 0.69$ | $7.78 \pm 1.17$ | $8.33 \pm 0.33$ |
| dodgeball | $7.33 \pm 7.57$ | $2.47 \pm 0.79$ | $2.8 \pm 1.44$ | $8.98 \pm 0.87$ |
| fruitbot | $8.0 \pm 2.65$ | $7.66 \pm 0.62$ | $7.19 \pm 1.09$ | $8.6 \pm 0.23$ |
| heist | $10.0 \pm 0.0$ | $0.0 \pm 0.0$ | $0.33 \pm 0.58$ | $9.11 \pm 1.02$ |
| leaper | $0.0 \pm 0.0$ | $0.0 \pm 0.0$ | $0.0 \pm 0.0$ | $0.0 \pm 0.0$ |
| maze | $10.0 \pm 0.0$ | $0.11 \pm 0.19$ | $5.56 \pm 5.09$ | $8.56 \pm 0.69$ |
| miner | $13.0 \pm 0.0$ | $0.94 \pm 0.48$ | $1.23 \pm 1.15$ | $11.37 \pm 0.23$ |
| starpilot | $18.0 \pm 10.54$ | $9.72 \pm 4.78$ | $12.9 \pm 4.69$ | $17.82 \pm 0.72$ |
| **Avgerage** | $6.96 \pm 1.25$ | $2.6 \pm 0.62$ | $3.81 \pm 0.68$ | $7.64 \pm 0.07$ |
| | **Human-normalized scores** | | | |
| bigfish | $0.09 \pm 0.09$ | $0.03 \pm 0.02$ | $0.04 \pm 0.0$ | $0.11 \pm 0.01$ |
| bossfight | $0.04 \pm 0.0$ | $-0.0 \pm 0.04$ | $0.03 \pm 0.02$ | $0.06 \pm 0.01$ |
| caveflyer | $-0.02 \pm 0.68$ | $-0.39 \pm 0.02$ | $-0.06 \pm 0.39$ | $0.73 \pm 0.0$ |
| chaser | $0.08 \pm 0.08$ | $0.1 \pm 0.04$ | $0.09 \pm 0.05$ | $0.18 \pm 0.04$ |
| coinrun | $0.33 \pm 1.15$ | $0.18 \pm 0.14$ | $0.56 \pm 0.23$ | $0.67 \pm 0.07$ |
| dodgeball | $0.33 \pm 0.43$ | $0.06 \pm 0.04$ | $0.07 \pm 0.08$ | $0.43 \pm 0.05$ |
| fruitbot | $0.28 \pm 0.08$ | $0.27 \pm 0.02$ | $0.26 \pm 0.03$ | $0.3 \pm 0.01$ |
| heist | $1.0 \pm 0.0$ | $-0.54 \pm 0.0$ | $-0.49 \pm 0.09$ | $0.86 \pm 0.16$ |
| leaper | $-0.43 \pm 0.0$ | $-0.43 \pm 0.0$ | $-0.43 \pm 0.0$ | $-0.43 \pm 0.0$ |
| maze | $1.0 \pm 0.0$ | $-0.98 \pm 0.04$ | $0.11 \pm 1.02$ | $0.71 \pm 0.14$ |
| miner | $1.0 \pm 0.0$ | $-0.05 \pm 0.04$ | $-0.02 \pm 0.1$ | $0.86 \pm 0.02$ |
| starpilot | $0.25 \pm 0.17$ | $0.12 \pm 0.08$ | $0.17 \pm 0.08$ | $0.25 \pm 0.01$ |
| **Average** | $0.33 \pm 0.14$ | $-0.14 \pm 0.03$ | $0.03 \pm 0.05$ | $0.39 \pm 0.0$ |

When reweighting by position, we assign $s_u(\tau_{\text{ret}}) = 1$ if $\tau_{\text{ret}}$ was generated before $\tau_{\text{in}}$ by the PPO agent that generated the data. Reweighting by position makes it likely that RA-DT observes the improvement steps in its context.

We find that task-based reweighting is essential for achieving the highest performance scores (see Figure 28). Using no reweighting at all results in a considerable drop in ICL performance. However, using retrieval with no task reweighting still compares favourably to uniform sampling across all

**Table 6:** Procgen Train Tasks, Evaluation Seeds.

| Environment | DT | AD | DPT | RA-DT |
|---|---|---|---|---|
| **Rewards** | | | | |
| bigfish | $0.0 \pm 0.0$ | $0.37 \pm 0.64$ | $0.04 \pm 0.08$ | $0.38 \pm 0.3$ |
| bossfight | $0.33 \pm 0.58$ | $0.02 \pm 0.02$ | $0.01 \pm 0.02$ | $0.02 \pm 0.04$ |
| caveflyer | $6.67 \pm 5.77$ | $3.67 \pm 2.08$ | $7.67 \pm 1.33$ | $9.89 \pm 0.19$ |
| chaser | $2.79 \pm 0.65$ | $2.1 \pm 0.67$ | $2.75 \pm 0.93$ | $5.17 \pm 0.58$ |
| coinrun | $10.0 \pm 0.0$ | $9.11 \pm 0.84$ | $9.89 \pm 0.19$ | $10.0 \pm 0.0$ |
| dodgeball | $0.0 \pm 0.0$ | $0.29 \pm 0.3$ | $0.0 \pm 0.0$ | $0.47 \pm 0.41$ |
| fruitbot | $5.0 \pm 4.0$ | $0.63 \pm 1.65$ | $1.04 \pm 0.83$ | $4.01 \pm 1.83$ |
| heist | $0.0 \pm 0.0$ | $0.0 \pm 0.0$ | $0.0 \pm 0.0$ | $0.11 \pm 0.19$ |
| leaper | $0.0 \pm 0.0$ | $0.11 \pm 0.19$ | $0.11 \pm 0.19$ | $0.22 \pm 0.38$ |
| maze | $6.67 \pm 5.77$ | $2.78 \pm 4.23$ | $1.67 \pm 2.89$ | $8.0 \pm 3.46$ |
| miner | $0.0 \pm 0.0$ | $0.58 \pm 0.31$ | $0.41 \pm 0.07$ | $0.77 \pm 0.09$ |
| starpilot | $16.0 \pm 1.0$ | $16.26 \pm 5.4$ | $15.81 \pm 3.27$ | $17.12 \pm 1.58$ |
| **Average** | $3.95 \pm 0.78$ | $2.99 \pm 0.92$ | $3.28 \pm 0.26$ | $4.68 \pm 0.33$ |
| **Human-normalized scores** | | | | |
| bigfish | $-0.03 \pm 0.0$ | $-0.02 \pm 0.02$ | $-0.02 \pm 0.0$ | $-0.02 \pm 0.01$ |
| bossfight | $-0.01 \pm 0.05$ | $-0.04 \pm 0.0$ | $-0.04 \pm 0.0$ | $-0.04 \pm 0.0$ |
| caveflyer | $0.37 \pm 0.68$ | $0.02 \pm 0.24$ | $0.49 \pm 0.16$ | $0.75 \pm 0.02$ |
| chaser | $0.18 \pm 0.05$ | $0.13 \pm 0.05$ | $0.18 \pm 0.07$ | $0.37 \pm 0.05$ |
| coinrun | $1.0 \pm 0.0$ | $0.82 \pm 0.17$ | $0.98 \pm 0.04$ | $1.0 \pm 0.0$ |
| dodgeball | $-0.09 \pm 0.0$ | $-0.07 \pm 0.02$ | $-0.09 \pm 0.0$ | $-0.06 \pm 0.02$ |
| fruitbot | $0.19 \pm 0.12$ | $0.06 \pm 0.05$ | $0.08 \pm 0.02$ | $0.16 \pm 0.05$ |
| heist | $-0.54 \pm 0.0$ | $-0.54 \pm 0.0$ | $-0.54 \pm 0.0$ | $-0.52 \pm 0.03$ |
| leaper | $-0.43 \pm 0.0$ | $-0.41 \pm 0.03$ | $-0.41 \pm 0.03$ | $-0.4 \pm 0.05$ |
| maze | $0.33 \pm 1.15$ | $-0.44 \pm 0.85$ | $-0.67 \pm 0.58$ | $0.6 \pm 0.69$ |
| miner | $-0.13 \pm 0.0$ | $-0.08 \pm 0.03$ | $-0.09 \pm 0.01$ | $-0.06 \pm 0.01$ |
| starpilot | $0.22 \pm 0.02$ | $0.22 \pm 0.09$ | $0.22 \pm 0.05$ | $0.24 \pm 0.03$ |
| **Average** | $0.09 \pm 0.1$ | $-0.03 \pm 0.1$ | $0.01 \pm 0.04$ | $0.17 \pm 0.05$ |

**Table 7:** Procgen Eval Envs.

| Environment | DT | AD | DPT | RA-DT |
|---|---|---|---|---|
| **Reward** | | | | |
| climber | $0.0 \pm 0.0$ | $0.0 \pm 0.0$ | $0.0 \pm 0.0$ | $0.0 \pm 0.0$ |
| ninja | $0.0 \pm 0.0$ | $0.0 \pm 0.0$ | $0.0 \pm 0.0$ | $1.89 \pm 2.71$ |
| plunder | $2.0 \pm 1.73$ | $0.27 \pm 0.13$ | $0.48 \pm 0.32$ | $2.39 \pm 0.67$ |
| jumper | $3.33 \pm 5.77$ | $2.78 \pm 2.83$ | $2.0 \pm 1.45$ | $4.33 \pm 2.33$ |
| **Average** | $1.33 \pm 1.01$ | $0.76 \pm 0.68$ | $0.62 \pm 0.37$ | $2.15 \pm 0.85$ |
| **Human-normalized Score** | | | | |
| climber | $-0.19 \pm 0.0$ | $-0.19 \pm 0.0$ | $-0.19 \pm 0.0$ | $-0.19 \pm 0.0$ |
| ninja | $-0.54 \pm 0.0$ | $-0.54 \pm 0.0$ | $-0.54 \pm 0.0$ | $-0.25 \pm 0.42$ |
| plunder | $-0.1 \pm 0.07$ | $-0.17 \pm 0.01$ | $-0.16 \pm 0.01$ | $-0.08 \pm 0.03$ |
| jumper | $0.05 \pm 0.82$ | $-0.03 \pm 0.4$ | $-0.14 \pm 0.21$ | $0.19 \pm 0.33$ |
| **Average** | $-0.19 \pm 0.19$ | $-0.23 \pm 0.1$ | $-0.26 \pm 0.05$ | $-0.08 \pm 0.12$ |

tasks. This result suggests that retrieval can play an important role in environments without a clear task separation or in scenarios where no task IDs are available.

In addition, we conduct a sensitivity analysis on the $\alpha$ parameter used in the re-weighting mechanism that determines how strongly the utility scores influences the final retrieval score. $\alpha = 1$ is used both during training for task-based reweighing and during evaluation for return-based reweighting (see Section 3). In Figure 29, we vary $\alpha$ **(a)** during training, or **(b)** during evaluation, while keeping the

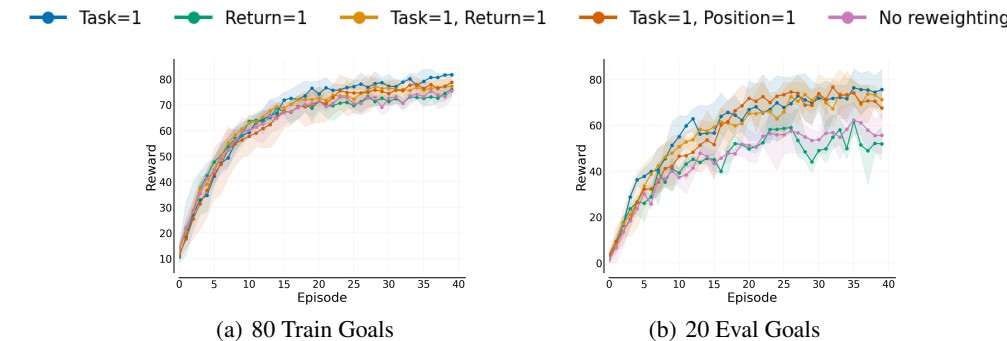

(a) 80 Train Goals      (b) 20 Eval Goals

**Figure 28:** Effect of the **Reweighting Mechanism**. Average performances on Dark-Room 10×10 over the course of training for **(a)** train and **(b)** test tasks.

other fixed. We find that RA-DT perform well for a range of values, but performance declines if no re-weighting is employed ($\alpha = 0$).

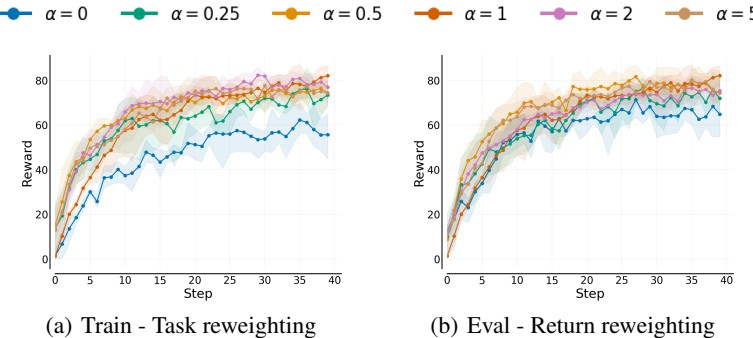

(a) Train - Task reweighting     (b) Eval - Return reweighting

**Figure 29: Sensitivity analysis** on $\alpha$ parameter used in re-weighting mechanism of RA-DT on Dark-Room 10×10.

### E.3 RETRIEVAL REGULARIZATION

Providing the agent with too similar trajectories, can encourage it to adopt copying behaviour instead of generating high-reward actions. To mitigate this, we found it useful to regularize the retrieval using three strategies: deduplication, similarity cut-off, and query dropout. To evaluate their individual impact on ICL performance, we systematically removed each one from RA-DT in Figure 30.

We find that deduplication plays the most significant role in enhancing performance. One reason, why deduplication is effective, is because RL datasets contain many very similar trajectories. Removing overlapping trajectories altogether is therefore beneficial for learning. Notably, deduplication also reduces the index size, thereby speeding-up the search process. The effect of deduplication may vary depending on dataset characteristics, such as state-action coverage [Schweighofer et al., 2022].

### E.4 QUERY CONSTRUCTION & SEQUENCE AGGREGATION

In RA-DT, we aggregate the hidden states of an input trajectory using mean aggregation of state tokens over the context length $C$ to obtain the $d_r$-dimensional query representation. It is, however, possible to use the hidden states of other tokens to construct the query. Therefore, we provide empirical evidence for this design choice in Figure 31a. We compare aggregating states, rewards, actions, returns-to-gos, all tokens, or only using the very last hidden state. Indeed, we find that aggregating state tokens gives the best results.

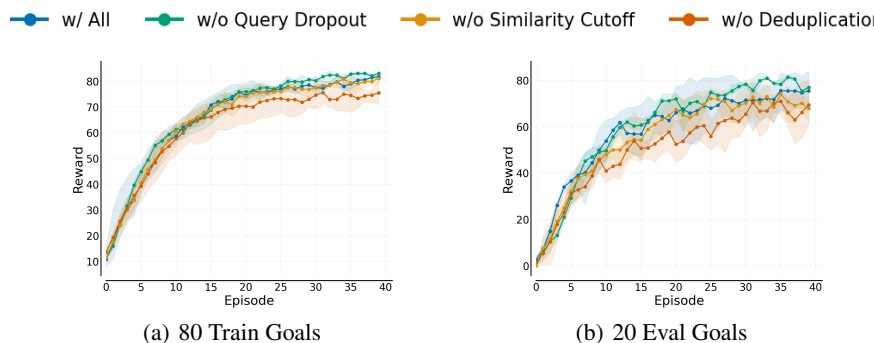

**Figure 30:** Effect of **Retrieval Regularization**. Average performances on Dark-Room $10\times10$ over the course of training for **(a)** train and **(b)** test tasks.

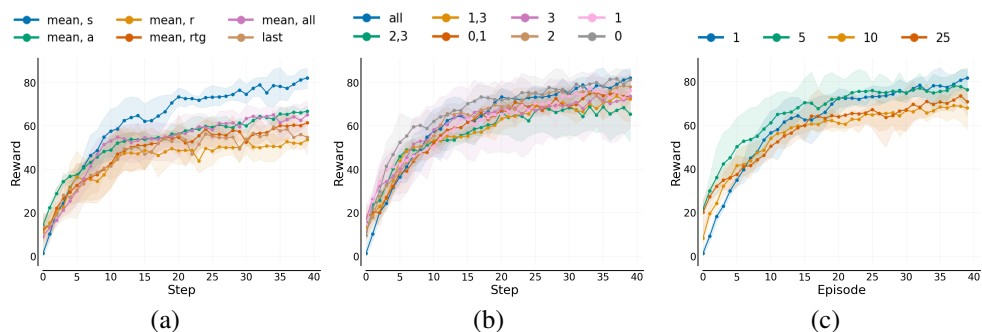

**Figure 31:** Ablations on important components of RA-DT conducted on Dark-Room $10\times10$. In **(a)** we investigate **sequence aggregations** to construct the query for retrieval. By default, we average state-tokens in the sequence (”mean, s”). In **(b)** we vary the **placement of cross-attention layers** in the DT. In **(c)** we vary the number of **steps in-between retrievals** during evaluation. We find that RA-DT delivers robust performance across settings.

### E.5 PLACEMENT OF CROSS-ATTENTION LAYERS

Next, we investigate the effect of the placement of the cross-attention layers in RA-DT. In Figure 31b, we therefore vary the placement of cross-attention layers in RA-DT. By default, we use cross-attention after every self-attention layer. We find that other choice also provide good results. While placing the cross-attention at bottom layers tends to be beneficial, placing them only upper level layers tends to hurt performance.

### E.6 INTERACTION STEPS BETWEEN CONTEXT RETRIEVAL

As mentioned in Section C.4, we perform context retrieval after every $t$ environment steps. Here, $t$ represents a trade-off between inference time and final performance. For grid-worlds, we use $t = 1$ by default. To better understand the effect of this design choice, we conduct an ablation in which we vary $t$ (see Figure 31c). Indeed, we find that higher values for $t$ result in a slight decrease in performance, but faster inference.

### E.7 EFFECT OF RETRIEVAL-AUGMENTATION ON TRAINING EFFICIENCY

Retrieval-augmentation adds computational overhead to the training pipeline due to the cost of embedding the query trajectories and searching for similar experiences in the vector index. Therefore, we study the effect of retrieval-augmentation on the training efficiency of RA-DT. For the purpose of this analysis, we measure training efficiency in terms of the *number of samples processed per second*

(higher is better). We run all experiments on an A100 GPU using the same training setup (batch sizes, context lengths) as described in Appendix C.

In Figure 32, we compare domain-specific/agnostic RA-DT to the three considered baselines on Dark-Room across gridsizes $10 \times 10$, $20 \times 20$, and $40 \times 20$. We find that the domain-specific variant of RA-DT attains minor training speed-ups on $10 \times 10$ and trains almost $7\times$ faster than baselines on the largest grid. The domain-agnostic variant of RA-DT, in contrast, exhibits slower training times on $10 \times 10$, but also trains significantly faster on the largest grid. Note that the differences among the three grid-sizes in the number of samples processed per second of RA-DT stem from the difference in sequence lengths ($C = 50$ for $10 \times 10$, $C = 100$ for $20 \times 20/40 \times 20$) and batch sizes ($B = 128$ for $10 \times 10/20 \times 20$, $B = 32$ for $40 \times 20$).

The efficiency gains of RA-DT are a direct result of the shorter required sequence lengths. In contrast to the baselines, the computational requirements of RA-DT do not grow with the episode length of the environment. Additional speed-ups can be achieved for RA-DT by pre-computing the retrieved trajectories prior to training similar to Borgeaud et al. [2022]. We also want to highlight that all baselines use FlashAttention to speed-up the training times and to ensure a fair comparison. Consequently, the empirical evidence demonstrates that RA-DT does not only improve the downstream performance in the environments, but is is also significantly faster to train (up to $7\times$).

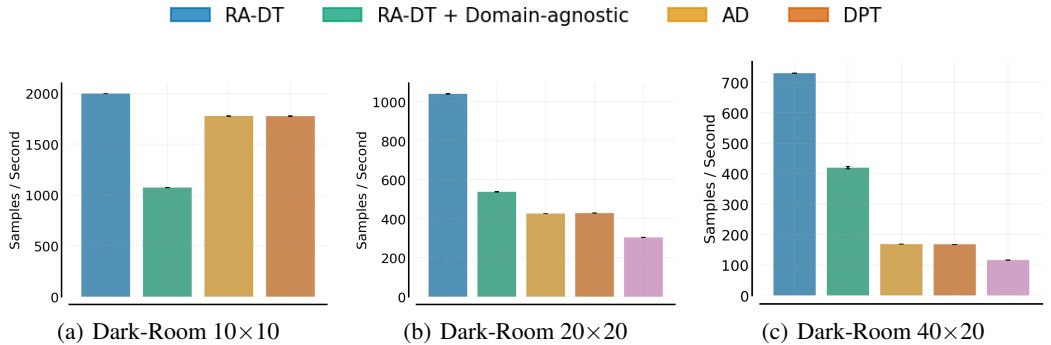

(a) Dark-Room $10\times10$      (b) Dark-Room $20\times20$      (c) Dark-Room $40\times20$

**Figure 32:** Training efficiency for all considered methods on **(a) Dark-Room 10×10**, **(b) Dark-Room 20×20**, **(c) Dark-Room 40×20**. We measure training efficiency in terms of the number of samples processed per second (higher is better). RA-DT achieves considerably speed-ups, in particular for larger grid-sizes.

### E.8    Effect of retrieval-augmentation on Inference efficiency

Retrieval-augmentation also incurs computational overhead during inference. Therefore, we study the effect of retrieval-augmentation on the infernce efficiency of RA-DT, similar to Appendix E.7. We measure inference efficiency in the *number of environment interaction steps performed per second* (higher is better). Note that this metric includes the environment latency. We average the inference efficiency metric across episodes to get a more robust estimate and discard the first episode to exclude compilation times. We conduct our analysis on an A100 GPU and use the same inference setup as described in Appendix C.

In Figure 33, we report the inference efficiency for domain-specific/agnostic RA-DT and the considered baselines on Dark-Room. For RA-DT, we report the inference times with $t \in \{1, 25\}$ where $t$ represents the number of interaction steps between retrievals. In Appendix E.6 we found that increasing $t$ only results in minor performance drops for RA-DT. For $t = 1$ RA-DT exhibits slightly slower inference speeds compared to the baselines. In contrast, for $t = 25$ there is no significant difference in inference speed between RA-DT and the baselines.

Note that the inference speed is roughly the same across grid sizes. This suggests that the inference time is not yet dominated by the quadratic cost of self-attention for $B = 1$ and the sequence lengths we consider in this analysis. To further support this, we run an ablation in which we compare the inference efficiency for all baselines with and without FlashAttention (see Figure 35) on Dark-Room $40 \times 20$. Indeed, we observe a significant drop in inference speed when FlashAttention is disabled.

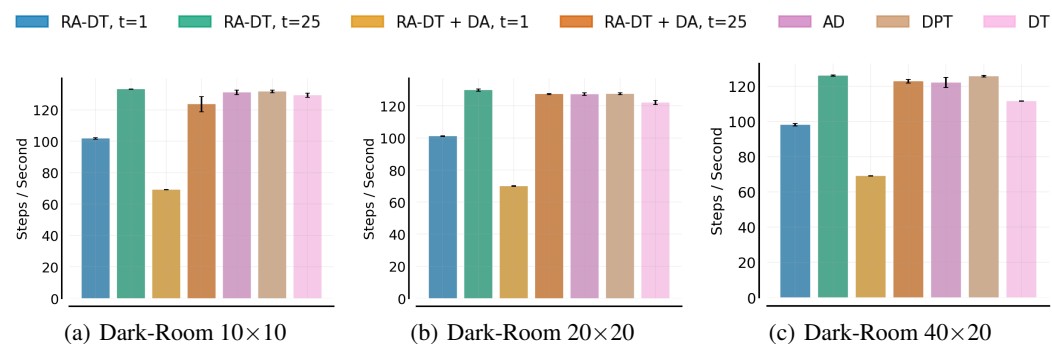

(a) Dark-Room 10×10        (b) Dark-Room 20×20        (c) Dark-Room 40×20

**Figure 33:** Inference efficiency for all considered methods on **(a) Dark-Room 10×10, (b) Dark-Room 20×20, (c) Dark-Room 40×20**. We measure inference efficiency in terms of the number of environment interaction steps performend per second (higher is better).

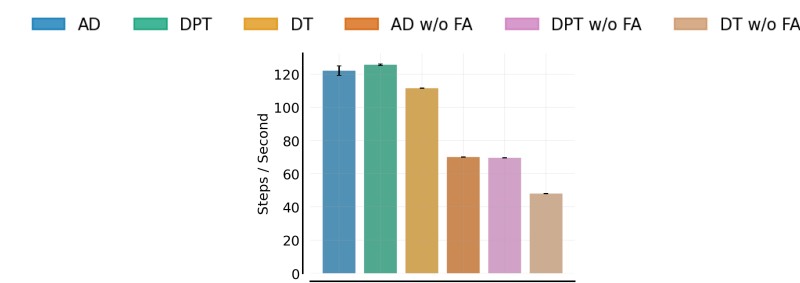

**Figure 34:** Effect of FlashAttention on inference efficiency of AD, DPT and DT on Dark-Room $40 \times 20$. Disabling FlashAttention results in a considerable drop in inference speed.

To conclude this analysis, our findings indicate that while RA-DT is slightly slower when retrieving on every step, it achieves comparable inference speeds to the baselines when retrieving less frequently. Importantly, the retrieval mechanism in RA-DT enables access to the entirety of the experiences collected across all ICL trials. In contrast, the baselines can only access experiences from a limit set of the most recent episodes that are preserved in the context (2 in our experiments). If we were to provide more context episodes to the baselines, the quadratic complexity of self-attention would kick in (similar to Figure 32). The ability of RA-DT to access a much broader set of experiences may be a reason for its enhanced down-stream performance.

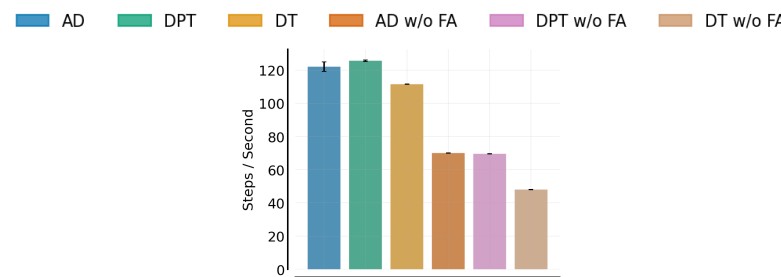

**Figure 35:** Effect of FlashAttention on inference efficiency of AD, DPT and DT on Dark-Room $40 \times 20$. Disabling FlashAttention results in a considerable drop in inference speed.

### E.9    PRE-TRAINED LANGUAGE MODEL

We investigate how strongly the ICL performance of RA-DT is influenced by the pre-trained LM used in our domain-agnostic embedding model. In Figure 36, we compare our default choice BERT

[Devlin et al., 2019] against four alternative encoder and decoder backbones, namely RoBERTa [Liu et al., 2019], DistilRoBERTa, DistilBERT [Sanh et al., 2019] and DistilGPT2. We find that RA-DT maintains decent performance across all pre-trained LMs, indicating robust retrieval performance across different LMs. Generally, the non-distilled variants outperform their distilled counterparts. Moreover, this experiment suggests a clear advantage of encoder-only models over the decoder-only LM, DistilGPT2. This suggests that the encoder-only LMs are better able to capture the relations between tokens within the token sequence, which leads to more precise retrieval of sub-trajectories and higher down-stream performance.

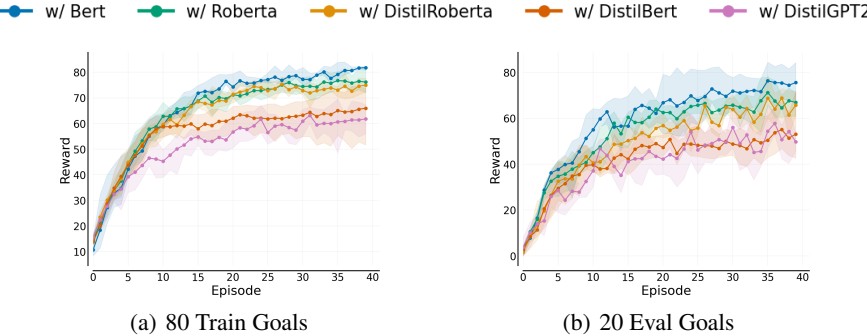

(a) 80 Train Goals  (b) 20 Eval Goals

**Figure 36:** Effect of the **Pre-trained LM**. Average performances on Dark-Room $10\times10$ over the course of training for **(a)** train and **(b)** test tasks.

### E.10  EFFECT OF $K$ ON ALGORITHM DISTILLATION

Finally, we investigate the effect of $K$ on the performance of AD. $K$ determines the number of episodes that have passed between the current and the context trajectory, which are provided to AD as the context. Consequently, $K$ specifies the extent of improvement observed between subsequent episodes. By default, we use $K = 100$ for our experiments on Dark-Room $10 \times 10$. Therefore, we conduct an ablation study, in which we very $K$ (see Figure 37. We find that too small values for $K$ (e.g., 1 and 10) result in slow ICL behavior. In contrast, too high values for $K$ (e.g., 500) lead to fast initial progress but suboptimal performance in the long term. Only $K = 100$ leads to steady improvement across all interaction episodes. Consequently, AD requires careful tuning of $K$.

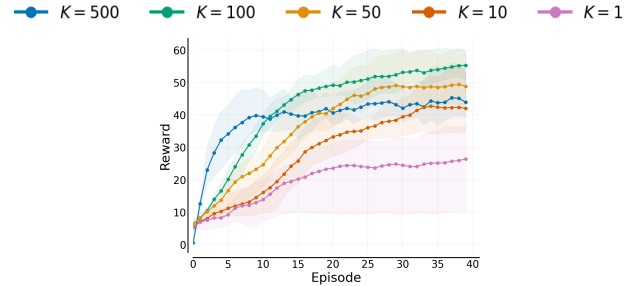

**Figure 37:** Ablation on the number of episodes $K$ in AD that have passed between "current" trajectory and "context" trajectory on Dark-Room $10\times10$. $K$ determines how much improvement is observed between episodes. We find that performance increases as $K$ increases, but only up to a certain point ($K = 100$). With $K = 500$, AD improves rapidly in the first few episodes, but then flattens out.

### E.11  CONVERGENCE OF BASELINES

In the main experiments on grid-worlds reported in Section 4.1, we found that the baselines AD and DPT only reach sub-optimal performance within the 40 ICL trials. Therefore, we analyse their

performance if evaluate for more ICL trials. In Figure 38, we compare the evaluation performance of AD and DPT across 200 ICL trials on the 20 hold-out tasks for Dark-Room $10 \times 10$. We find that both method continue to improve towards optimal performance in this environment when given more ICL trials. For this ablation, we found it useful to set $K = 50$ in AD (see Appendix E.10) instead of $K = 100$ as used in our main experiments over 40 ICL trials.

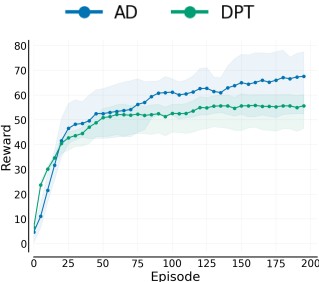

**Figure 38:** Evaluation of AD and DPT on Dark-Room $10 \times 10$ over 200 ICL trials. Both methods continue to improve towards optimal performance on this environment.

