# OpenReview forum: "Retrieval-Augmented Decision Transformer: External Memory for In-context RL"
_ICLR.cc/2025/Conference — Submitted to ICLR 2025_

### Official Review · Reviewer_Yuoy · 2024-10-26

**Soundness:** 3
**Presentation:** 2
**Contribution:** 1
**Rating:** 1
**Confidence:** 3

**Summary:**

This paper introduces Retrieval-Augmented Decision Transformer (RA-DT), a method that employs an external memory mechanism to store and retrieve relevant past experiences, enabling efficient in-context reinforcement learning in environments with long episodes and sparse rewards.

**Strengths:**

-Model agnostic Retrieval-Augmentation

-Smaller context length and boost in Performance

**Weaknesses:**

1) Novelty: The novelty of this work seems limited as it primarily leverages existing techniques like retrieval augmentation and Decision Transformers.

2) Qaulity Data Availability and Relevance: If quality relevant data is not available is in storage then it is not very helpful.

**Questions:**

1) Technical Challenges: Could you please elaborate on the specific technical challenges (that authors solve) in this work?  (Not including what prior works do). If Retrieval Augmentation and decision transformer already exist then what is the contribution?

2) Large Context Length and Retrieval Augmentation: How does your approach handle longer trajectories, and what are the potential limitations of using large context lengths in retrieval augmentation?

3) Domain-Agnostic Model Performance: In Figure 3b, why does the RA-DT + domain-agnostic model exhibit a performance decline compared to the RA-DT model?

4) In Figure 4, does running for more episodes ensure other approaches converge? Can you present those results to ensure baselines converge but require more samples?

5) Computational Efficiency: How does the computational cost of your approach compare to the baseline methods, especially in terms of inference time?

---

> ### Author Response · Authors · 2024-11-25
>
> We thank the reviewer for their feedback.
>
> **Novelty and Technical Challenges:** It is true that we are not the first to leverage retrieval-augmentation. However, there are a number of fundamental differences to existing works. The main challenge we address in this work is to improve in-context RL via retrieval-augmentation. To achieve this, we address the following technical challenges:
> - **Reliance on Long Context:** Existing ICRL works rely on keeping multiple episodes in their context, which is challenging as episodes in RL can last for thousands of timesteps. To address this, we leverage retrieval-augmentation to alleviate the need for long contexts that span multiple episodes. Importantly, the retrieval-augmentation in RA-DT enables access to the entirety of previously collected experiences at inference time with small additional cost (see below). This would be very computationally expensive for existing ICRL methods.
> - **Challenges of Retrieval in RL:** Targeting in-context RL with retrieval requires a number of careful technical considerations, on which we conduct a number of ablation studies (see Appendix E). Therefore, we leverage a reweighting mechanism (similar to [1]) to provide useful context trajectories at training and to produce high reward behavior at inference time. For this, we come up with measures for utility that are adequate for the RL setting, which we study in our ablation section (see Section 4.6, Appendix E.2). In addition, we found it useful to introduce retrieval regularization strategies (Appendix E.3).
> - **Domain-agnostic Retrieval:** To use retrieval-augmentation in RL, an embedding model is required. While embedding models are readily available for language, they are rarely available in RL. Therefore, we proposed to leverage the FrozenHopfield [2] mechanism to map RL “tokens” to the language space. This allows us to leverage frozen LMs as embedding models for RL trajectories, which enables retrieval without domain-specific pre-training. In contrast, existing works that focus on retrieval in RL (online RL [3] and the Go [4]) relied on domain-specific embedding models.
>
> Furthermore, our work aims to provide an **improved understanding of in-context RL**. In addition to the proposed method, we also provide new insights into how in-context learning behaves in offline RL. Previous works primarily considered grid-world environments for showcasing in-context RL. In contrast, we also evaluate more complex environments and find that ICL does not transfer to those domains. Consequently, we illuminate the challenges that current in-context RL methods face (see Section 5).
>
> **Quality of Data in Retrieval Buffer:** We want to clarify that we do not make use of any demonstrations during online evaluation in this work. During inference, the retrieval buffer is empty and entirely populated by the agent’s generated experience. Consequently, on grid worlds, the agents are able to benefit from (initially) suboptimal data generated in previous trials. Also, see Figure 17 for an example.

---

> > ### Author Response · Authors · 2024-11-25
> >
> > Regarding your **questions:**
> > 1. See discussion above.
> > 2. RA-DT is designed to reduce the reliance on long contexts that span multiple episodes, which other methods rely on. Consequently, retrieval-augmentation enables performance benefits without having to increase the sequence length and the computational requirements are not dependent on the episode length. However, we believe that providing longer context may benefit RA-DT similar to LMs that combine retrieval-augmentation with longer context [5].
> > 3. It is true that RA-DT domain agnostic performs slightly worse in terms of final performance on evaluation tasks in Figure 3b. We attribute this gap to slightly less precise retrieval performance of the domain-agnostic embedding model. However, note that while there is a gap to the domain-specific embedding model, the domain-agnostic one still outperforms all baselines.
> > 4. Thank you for raising this point. To verify that the baselines continue to improve, we run them with an increased number of 200 ICL trials on Dark-Room 10×10. Indeed, we find that the baselines continue to improve and approach optimal performance in this environment (in particular AD, see Appendix E.11).
> > 5. We thank the reviewer for raising the point on computational efficiency. We agree that a study on this would strongly benefit the paper. Indeed, retrieval-augmentation introduces additional computational overhead compared to methods that do not use retrieval. In contrast to other methods, however, RA-DT does not require keeping entire episodes in the context, which decouples the computational requirements from the episode length. In our updated manuscript, we report metrics for the training and inference efficiency for grid worlds with varying grid sizes to highlight this effect. We find that RA-DT is up to almost 7 times faster at training time (for the largest grid, see Appendix E.7). At inference-time, RA-DT is slightly slower compared to baselines when retrieving on every step, but exhibits similar inference speeds when retrieving less frequently (see Appendix E.8). In Appendix E.6, we validate that less frequent retrieval does not severely affect performance. Importantly, the retrieval mechanism in RA-DT enables access to the entirety of the experiences collected across all ICL trials with negligible additional cost. The ability of RA-DT to access a broader range of experiences may be a reason for its enhanced down-stream performance.
> >
> > We revised our manuscript, and believe it has become considerably better because of your suggestions. In case any questions remain, we would be happy to answer them! Also, we appeal to the reviewer to re-check their assessment, because based on the complaints raised, we believe 1 is unwarranted.
> >
> > [1] Generative agents: Interactive simulacra of human behavior, ArXiv 2023 \
> > [2] History compression via language models in reinforcement learning, ICML 2022 \
> > [3] Retrieval-augmented reinforcement learning, ICML 2022 \
> > [4] Large-scale retrieval for reinforcement learning, NeurIPS 2022 \
> > [5] ”Retrieval meets Long Context Large Language Models”, ICLR 2024

---

### Official Review · Reviewer_FDLc · 2024-10-30

**Soundness:** 1
**Presentation:** 2
**Contribution:** 1
**Rating:** 5
**Confidence:** 4

**Summary:**

The work introduces a variant of transformer-based offline RL by modifying the Decision Transformer architecture by an external memory component and associated retrieval mechanism. The retrieval-augmented decision transformers (RA-DT) retrieval mechanism enables retrieval of relevant sub-trajectories such that the in-context learning behavior of the DT is improved. The approach is evaluated on a variety of RL problems, though strong benefits of the proposed methods can only be shown on grid-world environments.

**Strengths:**

The retrieval-augmentation idea is compelling. It particularly shines in its ability of cutting down the context length that is required for in-context reinforcement learning.
Further, the idea of utilizing pre-trained language models for the domain-agnostic embedding is very interesting and could be useful in this type of in-context RL.
Lastly, the work evaluates the approach on a broad variety of problems including more toy-like problems to larger and more complex ones.

**Weaknesses:**

While the work provides a broad evaluation of the proposed method, the experiments highlight the shortcomings of the method. Often the proposed method does not achieve better in-context learning abilities, particularly on problems that are not grid-worlds. Due to these shortcomings the authors provide a longer discussion section on the potential shortcomings of offline in-context RL methods, though this discussion does not provide explanations or better understanding why the proposed method did not work. In short, this section leaves many open questions about RA-DT .
Similarly, the domain agnostic embedding idea is put in the spotlight as a big benefit. However, experiments with this embedding seem to only have been conducted in the simplest setting where RA-DT works and there it does not seem to be a generally good choice. Further investigation is needed to properly understand if the domain agnostic approach actually  is a good choice or if it was simply random chance that it worked for some of the simplest cases.
As one central aspect of this work is concerned with reducing the context size needed for action prediction, I would have expected (at least a short) discussion on the work of Melo, ICML 22 ("Transformers are Meta-Reinforcement Learners") as this approach uses a "working memory" of only the five most recent transitions, when learning to reinforcement learn with a transformer. While this work might not be in the exact ICL learning regime as the work under review, I believe that they are related enough that this work warrants further comparison. Particularly of interest should be that in Melos work longer horizons become detrimental for learning and the learned agent struggles to generalize to completely unseen settings. A followup work to Melos (Shala et al. 2024 ("Hierarchical Transformers are Efficient Meta-Reinforcement Learners")) aims at improving the ICL abilities of this approach by introducing a "cross-episodic" memory where not only the most recent transitions are part of the working memory, but snippets of the most recent episodes as well. While this increases the overall used context size again, it does not require full episodes in the context and shows superior ICL on completely new test environments.

**Questions:**

* How does RA-DT + domain-agnostic compare to the baselines and RA-DT in the complex environments?
* Why can the works on using Transformers for Meta-RL use much shorter context-sizes?
* Which of the shortcomings of ICRL does RA-DT suffer from (the most) and which of the provided experiments show this?

---

> ### Author Response · Authors · 2024-11-25
>
> Thank you for your helpful feedback. We are glad that you find retrieval-augmentation to be a compelling idea and the domain-agnostic embedding interesting.
>
> **Poor performance on non-gridworlds:** While RA-DT does outperform all baselines on grid-worlds, the reviewer is correct in stating that RA-DT does not achieve better in-context learning abilities on more complex environments such as Meta-World. RA-DT does outperform baselines in terms of final average performance on hold-out tasks in Meta-World and DMControl. However, note that, to the best of our knowledge, no other method is capable of solving these environments by improving in-context learning behaviour without leveraging expert demonstrations. This is the reason we added a longer discussion section on the shortcomings of existing ICRL methods, and directions we believe are critical for future ICRL research to study. To answer your question (Q3): we believe that RA-DT and other ICRL methods may suffer from all the listed shortcomings, and future algorithms could greatly benefit from improvements along the axes highlighted in the paper. We argue this detailed analysis is a contribution of the paper, as we hope will allow future research to focus on the outlined shortcomings that we identified.
>
> **Domain-agnostic embedding model:** It is true that we primarily compared the domain-agnostic embedding model on the grid world environments. However, across these environments, we observe similar performance trends for the domain-agnostic embedding model as for the domain-specific one. Considering that the domain-agnostic embedding model is pre-trained purely on text, we believe this is remarkable. Furthermore, the domain-agnostic embedding model reduces training cost, as the pre-trained LM is readily available.
> We did not conduct the comparison on more complex environments with the domain-agnostic embedding model in our initial version, because we observed that neither RA-DT nor any other baseline exhibits ICL behavior. Furthermore, the results by [1] strongly indicated that domain-agnostic LMs indeed transfer to more complex environments (such as Procgen). However, to validate this, we run domain-agnostic RA-DT on DMControl. We find that similar performance trends hold (see Figures 27 and 28). For our final version, we aim to add the domain-agnostic RA-DT curves for Meta-World and Procgen.
>
> **Discussion on Melo, ICML22:** Thank you for bringing up the paper by Melo [2]. We agree with the reviewer that this paper, as well as the follow-up [3], are important related works, and we added them to our discussion (Section 5) in our manuscript. However, there are considerable differences between our work (and related ICRL methods) and theirs. The most severe difference is that we consider an offline setting, in which the policy is learned from pre-collected trajectories via a supervised learning objective. In contrast, their work learns the policy via online interaction and proper RL objectives, which is more similar to online ICRL methods like Amago [4]. However, using RL objectives in the offline setting is not straightforward and requires to incorporate constraints (e.g., pessimism [5]). Consequently, we believe that an additional comparison against existing online methods is out-of-scope for this work. Nevertheless, we agree that online interaction and RL objectives (as used by [2,3]) can both be important components for ICL to emerge, which is a point we also raised in Section 5.

---

> > ### Author Response · Authors · 2024-11-25
> >
> > **Why do Transformers for Meta-RL [2] use shorter context sizes?** Because [2] and [3] primarily consider fully observable environments with dense rewards (Meta-World, HalfCheetah, Ant) which can be solved without a memory component. In contrast, in this work we also study partially observable environments with sparse rewards, such as Dark-Room or MazeRunner. For these environments, prior works such as Algorithm Distillation found it critical to rely on a multi-episodic context, which implies a long context. In contrast, RA-DT does not require a long multi-episodic context, which makes it more efficient at training time (up to almost 7 times, see Appendix E.7) and comparable to baselines at inference time (see Appendix E.8). Importantly, the retrieval mechanism in RA-DT enables access to the entirety of the experiences collected across all ICL trials, which may be a reason for its enhanced down-stream performance.
> >
> > We hope to have answered all your questions and believe our manuscript has improved substantially from your suggestions. If any questions remain, we would be happy to engage in further discussion.
> >
> > [1] “History compression via language models in reinforcement learning.” ICML 2022 \
> > [2] “Transformers are Meta-Reinforcement Learners”, ICML 2022 \
> > [3] “Hierarchical Transformers are Efficient Meta-Reinforcement Learners”, ArXiv 2024 \
> > [4] “AMAGO: Scalable In-Context Reinforcement Learning for Adaptive Agents”, ICLR 2024 \
> > [5] “Conservative Q-Learning for Offline Reinforcement Learning”, NeurIPS 2020 \
> > [6] “In-context Reinforcement Learning with Algorithm Distillation”, ICLR 2023

---

> > > ### Comment · Reviewer_FDLc · 2024-11-28
> > >
> > > Thank you for the thorough rebuttal. I have read the other reviews and responses as well. I am more positive towards the work now, but not yet fully convinced that it is ready for publication. Particularly, while discussion of shortcomings of ICL methods is interesting, I do not think that the work substantiates these shortcomings well enough. Still, I have increased my score to 5. I will keep an eye on the discussions and look forward to other reviewer responses.

---

### Official Review · Reviewer_6hs5 · 2024-11-02

**Soundness:** 3
**Presentation:** 4
**Contribution:** 3
**Rating:** 8
**Confidence:** 4

**Summary:**

This paper tackles the challenge of in-context learning in complex RL environments, where long episodes and sparse rewards make traditional approaches difficult to apply effectively. To address this, the authors introduce the Retrieval-Augmented Decision Transformer (RA-DT), which utilizes an external memory to store past experiences. This memory enables the selective retrieval of relevant sub-trajectories based on the current context, allowing the model to make informed decisions without relying on entire episodes. Through extensive experiments, the authors demonstrate the effectiveness of RA-DT and show its superior performance over baseline methods. Additionally, they release datasets for multiple environments to support future research in in-context RL.

**Strengths:**

1. The proposed RA-DT leverages an external memory mechanism to store and retrieve relevant sub-trajectories, effectively tackling the challenge of managing context in environments with long episodes and sparse rewards.
2. The authors conducted extensive experiments to showcase the method’s effectiveness and to highlight the contribution of each module.
3. The authors released datasets for multiple environments, offering valuable resources to support future research in in-context RL.

**Weaknesses:**

1. Although the authors explain the rationale for using each module in RA-DT and demonstrate their effectiveness through experiments, the proposed RA-DT appears to be a combination of various existing methods [1, 2, 3]. What specific innovations do each of these methods introduce compared to the original approaches?
2. The processes of searching forsimilar experiences and reweighting retrieved experiences introduce additional computational overhead, which is neither discussed in detail nor evaluated in the experiments. Could you discuss the computational overhead introduced by the search and reweighting processes, and how this compares to the baseline methods?

[1] History compression via language models in reinforcement learning.
[2] Retrieval-augmented multimodal language modeling.
[3] Generative agents: Interactive simulacra of human behavior.

I look forward to the authors' response and am open to further discussion.

**Questions:**

1. Could you provide information on the runtime and computational resources required for the proposed method compared to the baseline methods? (The same as the second point in weaknesses)

---

> ### Author Response · Authors · 2024-11-25
>
> Thank you for your constructive feedback. We appreciate your positive assessment and are glad that you consider our method effective and the datasets we release a valuable resource for future research.
>
> **Differences to prior works:** As the reviewer correctly identified, our work does build on other works [1,2,3] to enable retrieval for in-context RL. However, there are a number of fundamental differences to these works:
> - [1] introduces the FrozenHopfield mechanism and uses it to leverage pre-trained LMs as a history compression mechanism for online RL. In contrast, we consider the offline in-context RL setting and propose to use the FrozenHopfield mechanism for retrieval-augmentation. While in NLP pre-trained models are readily available to embed sequences, pre-trained models are rarely available in RL. Therefore, we leverage the FrozenHopfield mechanism to map RL tokens to the language space. This allows us to repurpose frozen LMs as embedding models for RL trajectories, and enables retrieval without domain-specific pre-training.  Furthermore, our focus is on offline in-context RL, whereas [1] focuses on online RL.
> - [2] leverages retrieval to enhance the generation abilities of multi-modal models for images and text. In contrast, we leverage retrieval-augmentation as an external memory mechanism to enhance the ICL abilities of RL agents. In addition to techniques from [2], we leverage deduplication of the retrieval index to regularize the retrieval process.
> - [3] conducts a large-scale experiment of simulated agents represented by language models that interact with one another. Furthermore, all inputs/outputs are defined via language. They leverage a retrieval mechanism to search for relevant experiences, which they reweight according to relevance and utility. Reweighting in the context of RL requires re-defining quantities such as utility due to the differences between the domains. Therefore, we use different reweighting terms to provide useful context trajectories at training time and to produce high reward behavior at inference time. For this, we come up with measures for utility that are adequate for the RL setting which we study in our ablation section (see Section 4.6, Appendix E.2)
>
> Note that the domains of these methods differ strongly from ours. Consequently, all the components we introduce in our setting require careful considerations that we investigate in our ablation studies (see Section 4.6, Appendix E).
>
> **Computational Overhead:** The reviewer is correct in stating that the process of searching for and reweighting experiences introduces additional computational overhead compared to methods that do not use retrieval. In contrast to other methods, however, RA-DT does not require keeping entire episodes in the context, which decouples the computational cost from the episode length. In our updated manuscript, we report the training times for grid worlds with varying grid sizes to highlight this effect. We find that RA-DT is up to almost 7 times faster at training time (for the largest grid, see Appendix E.7). At inference-time, RA-DT is slightly slower compared to baselines when retrieving on every step, but exhibits similar inference speeds when retrieving less frequently (see Appendix E.8). In Appendix E.6, we validate that less frequent retrieval does not severely affect performance. Importantly, the retrieval mechanism in RA-DT enables access to the entirety of the experiences collected across all ICL trials with negligible additional cost. The ability of RA-DT to access a broader range of experiences may be a reason for its enhanced down-stream performance.
>
> Our manuscript improved considerably by addressing and incorporating your comments: thank you.
>
> [1] “History compression via language models in reinforcement learning”, ICML 2022 \
> [2] “Retrieval-augmented multimodal language modeling”, ICML 2023 \
> [3] “Generative agents: Interactive simulacra of human behavior”, ArXiv 2023

---

### Author Response · Authors · 2024-11-25

Dear Reviewers,

We thank you for your helpful comments and constructive feedback! We addressed your question and comments in our individual responses. Based on your feedback, we performed additional experiments. We report our new results in the revised manuscript. These changes considerably improved our paper - thank you. In light of our responses and the new version of the paper, we appeal to the reviewers to re-check their assessments.

Kind Regards

---

### Meta-Review · Area_Chair_eaiN · 2024-12-19

**Metareview:**

This work presents strong results in gridworlds for the proposed retrieval-based ICL approach. However, the authors only present their results on gridworlds, while relegating their results on non-gridworld environments to the appendix. As pointed out by Reviewer FDLc, the benefits of their retrieval-based approach is not as clear in these settings. The paper does not provide any analysis to understand why the method falls weak in these non-gridworld settings. This is a lost opportunity for the paper to provide a clearer understanding of the proposed method, but instead the currently framing of results comes across as a bit of a cherrypick. I encourage the authors to resubmit an improved version of their work featuring deeper analysis into what makes their method work.

**Additional Comments On Reviewer Discussion:**

Reviewers FDLc and Yuoy both point out that this work does not provide sufficient explanations of their empirical results. These concerns were not sufficiently addressed in the authors' rebuttal.

---

### Decision · Program_Chairs · 2025-01-22

Reject